# Complementing $CO_2$ emission reduction by Solar Radiation Management might strongly enhance future welfare

Koen G. Helwegen[1], Claudia E. Wieners[2,3], Jason E. Frank[1], and Henk A. Dijkstra[2,3]

[1]Mathematical Institute; Utrecht University, Netherlands
[2]Institute for Marine and Atmospheric research, Utrecht; Utrecht University, Netherlands
[3]Centre for Complex Systems Studies, Utrecht; Utrecht University, Netherlands

**Correspondence:** Claudia E. Wieners (c.e.wieners@uu.nl)

**Abstract.** Solar Radiation Management (SRM) has been proposed as a means to reduce global warming in spite of high greenhouse gas concentrations and to lower the chance of warming-induced tipping points. However, SRM may cause economic damages, and its feasibility is still uncertain. To investigate the trade-off between these (economic) gains and damages, we incorporate SRM into a stochastic-dynamic integrated assessment model and perform the first rigorous cost-benefit analysis of sulphate-based SRM under uncertainty, treating warming-induced climate tipping and SRM failure as stochastic elements. We find that within our model SRM has the potential to greatly enhance future welfare and merits being taken seriously as a policy option. However, if only SRM and no $CO_2$ abatement is used, global warming is not stabilised and will exceed 2K. Therefore, even if successful, SRM can not replace but only complement $CO_2$ abatement. The optimal policy combines $CO_2$ abatement and modest SRM and succeeds in keeping global warming below 2K.

*Copyright statement.* TEXT

## 1 Introduction

Despite the Paris agreement target to keep global mean temperature change "well below 2K" in order to prevent "dangerous climate change" (UNFCCC, 2015), no decisive reduction of $CO_2$ emissions has yet taken place (Le Quéré et al., 2018). This has sparked renewed interest in the possibility of cooling the climate system by geoengineering (Crutzen, 2006). Among several suggested approaches, only Solar Radiation Management (SRM), i.e. reflecting part of the incoming solar radiation back into space, has the potential to offset the global mean temperature changes projected by 2100 (Keller et al., 2014).

Several SRM techniques have been proposed (Latham et al., 2008; Ahlm et al., 2017; Gabriel et al., 2017; Seneviratne et al., 2018), although for some of them it is yet unknown whether they will be effective in cooling the planet and whether they will be technically feasible. The scheme that is most likely to become ripe for employment in the near future is sulphate aerosol-based SRM (McClellan et al., 2010; Moriyama et al., 2017). The scheme involves injecting precursor gases such as $SO_2$ into the stratosphere. This leads to the formation of reflective sulphate aerosols in the lower stratosphere which increase the Earth's albedo and cause surface cooling. Such cooling – of about 0.4K over several years (Stowe et al., 1990; Thompson et al., 2009)

– was observed following the Pinatubo eruption (Stowe et al., 1990; Thompson et al., 2009; Stenchikov et al., 1998; Robock, 2000) which injected 8-10 Mt(S) (megatonnes of sulphur), mainly as $SO_2$, into the stratosphere (Ward, 2009). It is still uncertain whether SRM can completely eliminate future global warming. High aerosol concentrations lead to faster coagulation, which reduces albedo and accelerates deposition (Visioni et al., 2017). One study (Kleinschmitt et al., 2018) suggests that SRM cannot provide a stronger negative radiative forcing than $-2W/m^2$, while others find that sufficiently strong forcing can be achieved, albeit at very high injection rates (Niemeyer and Timmreck, 2015; Niemeyer and Schmidt, 2017).

The potential benefits of SRM are obvious: a reduction of global warming and warming-induced damages, and a reduced transition likelihood of temperature-related climate tipping points (Cai et al., 2016). However, SRM cannot avert all climate change (Kravitz et al., 2013). In particular, global mean precipitation is expected to decrease (Andrews et al., 2010; MacMartin and Kravitz, 2016), and the spatial precipitation patterns may change. Ocean acidification will continue unless atmospheric $CO_2$ concentrations are reduced (Tjiputra et al., 2015).

The implementation costs of sulphate SRM are estimated to be $2 - 10 \times 10^9 \$/Mt$ of injected gas (McClellan et al., 2010; Moriyama et al., 2017), which is modest compared to the world GDP of $80 \times 10^{12}\$$ (data for 2017, World Bank (2017)). For comparison, building and installing enough solar cells to meet global energy demand would, at current prices, cost about $2.5 \times 10^{14}\$$, although prices are decreasing rapidly (Cassedy and Grossman , 2017). However, apart from moral issues (Robock et al., 2009), sulphate SRM may have damaging effects on human health (Effiong and Neitzel, 2016) and the environment (Pitari et al., 2014; Ward, 2009) that are still poorly understood (Irvine et al., 2017). A sudden discontinuation of SRM will cause rapid warming ("termination shock") to levels dictated by greenhouse gas concentrations (Brovkin et al., 2009; Matthews and Caldeira, 2007), which could put more stress on ecosystems and societies than a gradual warming (Trisos et al., 2018).

At least two major uncertainties are of great importance for cost-benefit analysis of SRM: the possibility of warming-induced tipping behaviour (whose likelihood is reduced by SRM) and the possibility of SRM failure, either by inefficiency (Kleinschmitt et al., 2018) or because (unforseen) damaging side-effects force one to abandon it (Robock et al., 2009). In this study we use a stochastic version of the integrated assessment model DICE (Nordhaus, 1992) to compute the (economically) optimal policy including $CO_2$ abatement and SRM.

Here we build on earlier studies, which often included uncertainty only through parameter sensitivity analysis (Goes et al., 2011; Bahn et al., 2015) or as a simplified two-step decision problem (Moreno-Cruz and Keith, 2013). Two recent studies (Heutel et al., 2016, 2018) include climate tipping behaviour and parameter uncertainty in DICE but employ a simple 4-step look-ahead scheme that is unsuitable for long-term optimisation. We employ dynamic programming (Bellman, 1957) to perform the first rigorous cost-benefit analysis of SRM under uncertainty, albeit with a simple model.

The DICE model has been criticised for being overly simple (Pindyck, 2017). In particular, it employs a very aggregated damage function for assessing the material and immaterial cost of climate change (see Nordhaus and Boyer (2000) for the calibration), which ignores irreversibility of damages and delayed damage (e.g. slow melt of ice caps) and which in later model versions (Nordhaus, 2018) has only received minor updates (Auffhammer, 2018), despite new studies on the subject (IPCC WG2, 2014). Neither does it include climate adaptation. In addition, DICE has an overly simplified energy sector with exogenous costs for CO2 reduction and does not include negative emission techniques. Finally, assuming only one global

"social planner", it disregards the possibility of conflict or imperfect collaboration. Despite these shortcomings, we believe DICE to be a useful testbed for exploratory studies, which should serve as a first orientation and be expanded using more detailed models.

The paper is organised as follows: In Sect. 2.1 we present our model GeoDICE, a stochastic DICE model including Geo-engineering, and in Sect. 2.3, we describe the scenarios employed. The results are presented in Sect. 3.1 (deterministic cases) and Sect. 3.2 - 3.3 (stochastic cases), with a sensitivity analysis in Sect. 3.5. A summary and discussion is presented in Sect. 4.

## 2   Methods

### 2.1   GeoDICE: a stochastic DICE model including Geoengineering

Our code is based on Cai et al. (2016), which in turn combines the 2013 version of DICE (Nordhaus, 1992, 2018) and the DSICE framework (Cai et al., 2012) for stochastic treatment of DICE. Here, we include SRM as an additional policy option (together with $CO_2$ abatement). To this end, we incorporate the cooling effect, implementation costs, and environmental damages of SRM into DSICE. A summary of the model parameters and their standard values is given in Table 1.

#### 2.1.1   SRM and Radiative Forcing

To the radiative forcing equation, we add a contribution that depends sublinearly on the sulphur injection rate (Niemeyer and Timmreck, 2015). The total radiative forcing $F$ takes the form

$$F = \alpha_{CO2} \ln((C_{PI} + C)/C_{PI}) + F_{other} - \eta\, \alpha_{SO2} \times \exp[-(\beta_{SO2}/I_S)^{\gamma_{SO2}}] \equiv F_C + F_{other} + F_S. \tag{1}$$

The first term $F_C$ describes the contribution of the increase $C$ in atmospheric $CO_2$ concentration above the preindustrial value $C_{PI}$ and is the same as in DICE. The second term $F_{other}$ represents the effects of other greenhouse gases (eg. $CH_4$, $N_2O$, halogen compounds) and industrial aerosol. In DICE, this term is prescribed. However, it seems unlikely that a society that makes great efforts towards abating $CO_2$ emissions does nothing towards combatting other pollutants.

Ciais et al. (2013) quantify various forcing agents, some of which we believe can be more easily abated than others. In particular, we assume that roughly 30% of the current $CH_4$ emissions (contributions related to fossil fuel production, e.g. leakage, and biomass burning), 10% of the $N_2O$ emissions (likewise from industry and fossil fuel production) and 100% of the emission of halogen compounds could be abated, whereas 70% of the $CH_4$ emissions (natural sources and agriculture) and 90% of $N_2O$ emissions (agriculture) cannot be abated. Tropospheric ozone, another important greenhouse gas, is formed in chemical reactions with pollutants which likewise can be partially abated. As a rough estimate, about 50% of the radiative forcing stemming from non-$CO_2$ greenhouse gasses could be abated. For simplicity, we assume that this also holds for (mainly cooling) industrial aerosol. Thus we put

$$F_{other} = F_{other,DICE} \times (1 - \kappa\mu),$$

| Symbol | Meaning | Value |
|--------|---------|-------|
| $\alpha_{CO2}$ | Effect $CO_2$ on radiative forcing; see eq. 1 | $5.35 W/m^2$ |
| $\alpha_{SO2}$ | scales sulphate radiative forcing; see eq. 1 | $65 W/m^2$ |
| $\beta_{SO2}$ | scales sulphate radiative forcing; see eq. 1 | $2246 Mt(S)/yr$ |
| $\gamma_{SO2}$ | scales sulphate radiative forcing; see eq. 1 | $0.23$ |
| $\eta$ | sulphate rad. forcing correction; see eq. 1 | $0.742$ |
| $b_1$ | strength temperature response; see eq. 4 | $0.126 \frac{K}{W/m^2}$ |
| $b_2$ | strength temperature response; see eq. 4 | $0.0254 \frac{K}{W/m^2}$ |
| $\tau_{T1}$ | time scale temperature response; see eq. 4 | $1.89 yr$ |
| $\tau_{T2}$ | time scale temperature response; see eq. 4 | $13.6 yr$ |
| $p_T$ | precipitation dependence on temp.; see eq. 6 | $0.0806 (mm/day)/K$ |
| $p_C$ | precipitation dependence on $CO_2$; see eq. 6 | $-0.0229 (mm/day)/(W/m^2)$ |
| $p_S$ | precipitation dependence on SRM; see eq. 6 | $-0.0077 (mm/day)/(W/m^2)$ |
| $\psi_C$ | econ. damage from $CO_2$ conc. ; see eq. 9 | $1.703 \times 10^{-3} K^{-2}$ |
| $\psi_T$ | econ. damage from warming; see eq. 9 | $0.4 (mm/day)^{-2}$ |
| $\psi_P$ | econ. damage from precip. change; see eq. 9 | $3.31 \times 10^{-8} (ppmv)^{-2}$ |
| $\psi_S$ | econ. damage from SRM; see eq. 9 | $9.27 \times 10^{-5} (Mt(S))^{-2}$ |
| $\psi_{fail}$ | econ. damage from SRM failure; see sect. 3.3 | $0.01/Mt(S)$ |
| $\Omega$ | remaining fraction econ. output after tipping; see eq. 7 | $0.9$ |
| $\lambda_S$ | implementation cost SRM; see eq. 7 | $14 \times 10^9 \$/Mt(S)$ |
| $\lambda_0$ | cost of abatement; see eq. 8 | $2.15$ |
| $\lambda_1$ | cost of abatement; see eq. 8 | $0.418\$/kg(C)$ |
| $\lambda_2$ | cost of abatement; see eq. 8 | $2.0$ |
| $\lambda_3$ | cost of abatement; see eq. 8 | $0.005 yr^{-1}$ |
| $\kappa_{tipp}$ | tipping probability per year and K warming; see eq. 10 | $0.00255/yr/K$ |
| $T_{tipp}$ | temperature threshold for (damage) tipping; see eq. 10 | $2K$ |
| $T_{alb}$ | temperature threshold for albedo tipping; see eq. 11 | $1.5K$ |
| $\alpha_{alb}$ | radiative forcing strength for albedo tipping; see eq. 11 | $1.07 W/m^2/K$ |
| $\kappa_{fail}$ | probability of SRM failure per year; see sect. 2.1.4 | $0.00056/yr$ |
| $\rho$ | pure rate of time preference (constant in time); see eq. 12 | $0.015/yr$ |
| $\delta_K$ | rate of capital depreciation; see Nordhaus (1992) | $0.065/yr$ |

**Table 1.** Model parameters of the GeoDICE model related to the representation of SRM. The carbon model parameters can be found in Table 5 of Joos et al. (2013), and others in DICE/DSICE (Cai et al., 2012).

where $\kappa = 0.5$, $\mu$ is the abatement of $CO_2$, i.e. the fraction of $CO_2$ emissions avoided, and $F_{other,DICE}$ is the prescribed contribution used in DICE.

The third term $F_S$ describes the influence of sulphate SRM. The sulphate injection leads to a (negative) radiative forcing at the top of the atmosphere which is given by $\alpha_{SO2} \times \exp[(-\beta_{SO2}/I_S)^{\gamma_{SO2}}]$, as found in an atmospheric chemistry modelling study (Niemeyer and Timmreck, 2015), where $I_S$ is the annual injection rate of sulphur into the stratosphere (measured in Mt(S)/year).To achieve a modest radiative forcing of $-2W/m^2$ at the Top Of the Atmosphere (TOA), an annual injection of 10 Megatonnes of sulphur (Mt(S)), equivalent to one Pinatubo eruption, is required, whereas to achieve a forcing of $-8.5W/m^2$ (offsetting the greenhouse gas forcing projected for 2100 under the RCP8.5 scenario), 100Mt(S)/year is needed. However, due to fast adjustment processes, the TOA radiative forcing is not sufficient to predict the impact on global mean surface temperature (Kravitz et al., 2015). For example, it is found (MacMartin and Kravitz, 2016) that to compensate $7.42W/m^2$ forcing from quadrupling $CO_2$, the solar constant would have to be reduced by $(4.2 \pm 0.6)\%$, which amounts to $10.1W/m^2$ TOA (taking into account the Earth's albedo). In other words, the top of atmosphere radiative forcing arising from changes in the solar constant is less efficient than forcing caused by $CO_2$ by a factor of $\eta = 0.742$. We assume here that sulphate SRM has the same efficiency factor $\eta$ as solar dimming, since both processes take place above the troposphere, and multiply the sulphate SRM contribution to $F$ by this factor in eq. (1). Note that there is still considerable uncertainty about the forcing efficiency of SRM. For example, Tilmes et al. (2018) find higher efficiencies and an almost linear relationship for injection rates up to $25Mt(S)/yr$, while Kleinschmitt et al. (2018) suggests that the maximal radiative forcing achievable with sulphate SRM might be limited to $2W/m^2$. This possibility that SRM is much less efficient is qualitatively included in the Realistic Storyline scenario described below, which captures that SRM may never work at all. For numerical reasons, we impose an upper bound of $I_S \leq 100Mt(S)/yr$ on the injection rates, i.e. we do not allow them to exceed $\approx 10$ Pinatubo eruptions per year. This upper limit is a much higher injection rate than considered in most detailed studies of the environmental and climate effects of SRM. The limit is never reached except in the somewhat extreme SRM-only scenario (see sect. 2.3).

### 2.1.2 Carbon cycle and Climate response

We replace the carbon-climate part of DSICE by an emulator of full-fledged climate model simulations (Aengenheyster et al., 2018; MacMartin and Kravitz, 2016). We also include global mean precipitation as a proxy for the residual climate change (changes remaining if SRM is employed to keep global mean temperature constant).

As in DICE, $CO_2$ can be emitted by fossil fuel combustion and landuse change. The former contribution is calculated within our model, the latter is prescribed externally, using the same values as DICE. We model carbon concentrations based on the Green's function found by Joos et al. (2013).

Current $CO_2$ concentrations $C(t)$ (above pre-industrial) can be computed from emissions $E$ at all previous times $t' < t$:

$$C(t) = \int\limits_{t' < t} G_C(t - t')E(t')dt', \tag{2}$$

where $G(t - t')$ is the Green's function determining how a unit emission pulse contributes to the concentrations $t - t'$ years later, and $E(t')$ an emission pulse at time $t'$. Following Joos et al. (2013), $G_C(s)$ can be represented as a sum of exponentials,

$G_C(s) = a_0 + \sum_{n=1}^{N} a_s e^{-s/\tau_n}$, with $n = 3$, and the temporal evolution of $C$ can be rewritten as

$$C(t) = C_0(t) + \sum_{n=1}^{N} C_n(t), \tag{3a}$$

$$dC_0/dt = a_0 E, \tag{3b}$$

$$dC_n/dt = a_n E - \frac{1}{\tau_n} C_n. \tag{3c}$$

Here $a_0$ represents the fraction of carbon emissions staying permanently in the atmosphere. The model parameters $a_n, \tau_n$ were obtained from a multi-model study (Joos et al. (2013), see their Table 5) and thus represent a best estimate for the behaviour of the carbon cycle, provided that nonlinear effects (e.g. saturation of carbon sinks with increasing $CO_2$ concentrations) are small. The initial values are $C_0 = 39.01 ppmv$, $C_1 = 35.84 ppmv$, $C_2 = 21.74 ppmv$, $C_3 = 4.14 ppmv$, since our model does not start at pre-industrial times (1765) but in 2005.

For the global mean temperature change (relative to pre-industrial), we follow the same approach, fitting the temperature response to a 1-year pulse of radiative forcing obtained by a multi-model study (MacMartin and Kravitz, 2016) onto a sum of exponentials, obtaining

$$T = T_1 + T_2 \tag{4a}$$

$$dT_n(t)/dt = b_n F - \frac{1}{\tau_{Tn}} T_n \tag{4b}$$

where $F$ is the radiative forcing from eq. (1) and other parameters are in Table 1. For the temperature response to a radiative forcing pulse, there is no permanent response $T_0$. The initial values (year 2005) are $T_1 = 0.466K$ and $T_2 = 0.436K$.

The response of global mean precipitation $P$ to $CO_2$-induced or SRM-induced radiative forcing is based on MacMartin and Kravitz (2016) and can be split into a slower temperature-driven increase of $2.5\%/K$ and an instantaneous contribution due to $CO_2$ and SRM (Andrews et al., 2010). In particular, increased $CO_2$ concentrations cause additional absorption of longwave radiation, warming the atmosphere and causing a more stable stratification, which suppresses precipitation, while surface warming enhances precipitation. For a gradual increase in $CO_2$ and zero SRM, the temperature-driven effect dominates over the instantaneous contribution, leading to a net moistening. For SRM, the instantaneous contribution is much weaker than for $CO_2$. More specifically, the response $G_P^f$ of the global mean precipitation $P$ to a one-year-long $1W/m^2$ pulse of radiative forcing from agent $f$ ($f$ stands for $CO_2$ or a change in the solar constant) in year 0, obtained by MacMartin and Kravitz (2016), can be expressed as

$$G_P^f(t) = b_f \delta_{t,0} + a G_T(t), \tag{5}$$

where $G_T$ is the temperature response to a $1W/m^2$ forcing and $\delta_{t,0} = 1$ if $t = 0$ and 0 else. This means that in the year of the forcing pulse, the fast response $b_f$ plays a role, whereas in later years, the precipitation response is determined by the temperature response. As before, we use the result for reduction in the solar constant as a proxy for sulphate SRM. By lack of data, the fast response to other forcing agents constituting $F_{other}$ is ignored. With these results, the change in global mean

precipitation w.r.t pre-industrial can be written as

$$P(t) = p_T T(t) + p_C F_C(t) + p_S F_S(t). \tag{6}$$

where $p_T T$ is the 'slow' precipitation change mediated by warming, whereas $p_C F_C$ and $p_S F_S$ are the instantaneous responses, expressed in terms of the radiative forcings $F$. Throughout our study, $F_C > 0$ ($CO_2$ leads to a positive radiative forcing) and $F_S < 0$ (SRM is used to lower the radiative forcing). As explained above, $p_T > 0$ and $p_C < p_S < 0$. Therefore, if SRM were employed such as to cancel the global mean temperature change ($F_S = -F_C - F_{other}$, hence $T = 0$), the slow responses stemming from temperature change would cancel and the fast response to $CO_2$ would dominate, reducing $P$.

We use $P$ as a proxy for residual climate change, i.e. for all effects which remain even if global mean temperature changes are cancelled by SRM.

### 2.1.3 The Damage function and SRM costs

As in DICE (Nordhaus, 1992), the gross domestic product (GDP) $\bar{Y}$ is diminished by climate-related damage and by expenditures for climate policy ($CO_2$ abatement and SRM implementation). Including these losses, we retain for the net output:

$$Y = \Omega \frac{1}{1+D} \Lambda \bar{Y} - \lambda_S I_S. \tag{7}$$

Here, $\Omega$ describes damage due to tipping points (see sect. 2.1.4). If tipping has occurred, then $\Omega = 0.9$ (reducing the economic output), else $\Omega = 1$ (output not reduced). $D \geq 0$ describes non-tipping damage (discussed below). $\Lambda$ is a factor describing the abatement costs ($\Lambda < 1$ in case of abatement, and $\Lambda = 1$ in case of no abatement) taken over from DICE-2013 (Nordhaus and Boyer, 2000; Nordhaus, 2018):

$$\Lambda(\mu) = 1 - \Lambda_0(t)\mu^{\lambda_0} \tag{8a}$$

$$\Lambda_0(t) = \frac{\lambda_1 \sigma(t)}{\lambda_2}[\lambda_2 - 1 + \exp(-\lambda_3 t)] \tag{8b}$$

where $\sigma(t)$ is the carbon intensity (amount of carbon released per dollar production, in absence of abatement), and $\lambda_i$ are constants. Since $CO_2$ emission is proportional to $\bar{Y}$, so are abatement costs (the more economic output, the more $CO_2$ emissions and hence the higher the costs of eliminating a fraction $\mu$ of these emissions). $\lambda_S I_S$ is the implementation cost of SRM, which we assume to be linear in the injection rate $I_S$ and independent of $\bar{Y}$. Two studies (McClellan et al., 2010; Moriyama et al., 2017) suggest that the costs for lifting gasses to 20km height are of the order of $2 - 10 \times 10^9 \$/Mt$ of injected gas. Taking an intermediate value of $7 \times 10^9 \$/Mt$, and assuming that the gas used is $SO_2$ (which has twice the molecular weight of elementary S), this amounts to $14 \times 10^9 \$/Mt(S)$. Note that $H_2S$ would have a lower weight per mole of S, which might reduce transportation cost. However, $H_2S$ is also more poisonous and thus potentially harder to handle. To be conservative, we assumed the costlier solution.

While the original DICE model assumes that climate-induced damage $D$ scales with the square of temperature change $T$ ($D(T) = \psi_0 T^2$ with constant $\psi_0$), we keep the quadratic structure but split the damage function into three climate-related

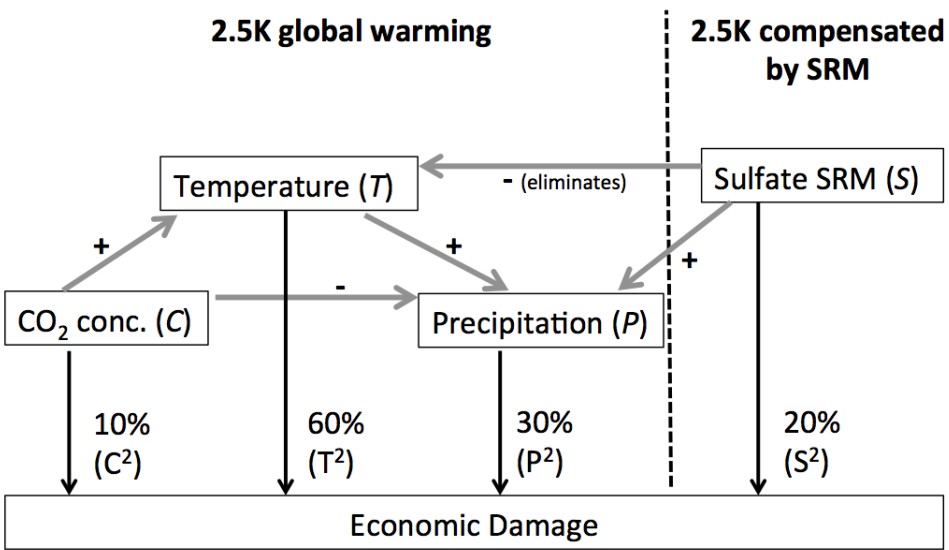

**Figure 1.** Graphical representation of the damage function. The black thin arrows represent contributions to the damage function, while grey arrows depict how the climate variables influence each other (+ and - stand for increasing and decreasing effects, respectively; see eq. (4) and (6)). The percentages for $C$, $T$, and $P$ are based on the contributions of these variables for the standard case of 2.5K warming in equilibrium and in absence of SRM. In this standard case, our damage equals that of the DSICE model (Cai et al., 2012). The sulphur injections that would be needed to offset 2.5K warming cause a direct damage of 20% of the standard damage function.

contributions and one contribution representing the damage inflicted by sulphate SRM (see fig. 1):

$$D(T,P,C,I_S) = \psi_T T^2 + \psi_P P^2 + \psi_C C^2 + \psi_S I_S^2 \qquad (9)$$

where $T$, $P$, $C$ are the changes (w.r.t. pre-industrial) of global mean temperature, global mean precipitation and atmospheric $CO_2$ concentration, respectively, and $I_S$ the sulphur injection rate in $Mt(S)/yr$. Note that while SRM counteracts the effect of $CO_2$ on both temperature and precipitation, the relative influence of the forcing agents on both variables differs, so that it is not possible to compensate the warming and precipitation change at the same time. Both positive and negative precipitation changes $P$ are considered damaging, because both require ecosystems and humanity to adapt. An increase in atmospheric $CO_2$ may not be damaging in itself, or even benefit plant growth (Ciais et al., 2013); but we consider $C$ as a (rough) proxy for ocean acidification, which we do not model explicitly. The coefficients $\psi$ (values: see Table 1) are chosen such that for the standard case of 2.5K warming in equilibrium without SRM, our total damage equals that of the original DICE model, and the contributions by $T$, $P$, and $C$ are 60%, 30%, and 10%, respectively. The standard case was determined by running the climate module of GeoDICE to equilibrium with constant greenhouse gas concentrations, such as to obtain $T = 2.5K$. Following Aengenheyster et al. (2018), and approximately in agreement with RCP scenarios, it was hereby assumed that other forcing agents (other greenhouse gases and aerosols constituting $F_{other}$) contribute 14% of the total radiative forcing. These

other forcing agents are not assumed to cause direct damage. The damages associated with the annual sulphur injections needed to offset a warming of 2.5K are assumed to equal 20% of the standard case damage.

Previous studies (Heutel et al., 2016, 2018) likewise split the damage function, but without including residual climate change ($P$). Heutel et al. (2018) assume oceanic and atmospheric $CO_2$ to cause 10% of the the total damage each (20% in total). However, atmospheric $CO_2$ is not known to cause substantial direct damage and may even be beneficial to plant growth (Ciais et al., 2013), while oceanic $CO_2$ leads to ocean acidification. As mentioned, we do not explicitly compute oceanic $CO_2$, but reduce total $CO_2$-related damage to 10% because half of the total damage in Heutel et al. (2018) seems in fact small. The splitting between $T$ and $P$ is somewhat arbitrary, but is based on the rough assumption that, although precipitation changes can have substantial impact, much of the damage is either determined by temperature (especially sea level rise, a major contributor) or at least strongly influenced by it (e.g. hurricanes), hence $\psi_T > \psi_P$. The damage related to SRM depends on the injection rate, not on the percentage of compensated greenhouse gas forcing as was (somewhat unrealistically) assumed in earlier studies (Heutel et al., 2016, 2018). The choice of $\psi_S$ is again somewhat arbitrary, as virtually no data on the economic damage of SRM is available. However, our main conclusions are unaffected by the exact choice of the parameters $\psi$ (see Sect. 3.5).

### 2.1.4 Tipping points and SRM failure

Climate change may not only lead to smooth and predictable damages, but also induce low-probability, high-impact, irreversible events such as a collapse of ice sheets (Cai et al., 2015, 2016). The chance of such tipping behaviour is thought to increase with temperature. We take tipping into account in a stylised way, assuming that there is one tipping event that, once activated, reduces GDP by 10% for all subsequent time steps (i.e. $\Omega = 0.9$ in eq. (7)). The likelihood of tipping obeys

$$L_{tipp} = \begin{cases} 0 & T < T_{tipp} \\ (T - T_{tipp}) \times \kappa_{tipp} & T > T_{tipp} \end{cases}, \tag{10}$$

i.e., it is zero if the global mean temperature change $T < T_{tipp} = 2K$, but increases linearly with warming above 2K. While in the real climate system a sharp threshold might not exist, this choice reflects 'political reality', in which policy makers set thresholds for 'dangerous' climate change to be avoided. The constant $\kappa_{tipp}$ is chosen such that in a scenario where the policy maker uses only abatement and remains unaware of possible tipping behaviour, the probability of tipping within 400 years is 50%. This order of magnitude of the likelihood and damage of tipping is consistent with earlier studies (Cai et al., 2016).

We also take into account the possibility that SRM has to be abolished. While possible reasons remain speculative at this point, it is not inconceivable that SRM has an unexpected destructive side effect, such as a massive deterioration of the ozone layer. We model this by assuming that each year, there is a probability $\kappa_{fail}$ that SRM may not be applied anymore in the future. The cumulative probability of SRM failure over 400 years is 20%. Failure is assumed irrevocable; once failed, SRM remains unavailable forever. In the basic scenarios (see sect. 2.3), we include no economic damage related to SRM failure, because humanity is optimistically assumed to realise such dangers and abandon SRM in time (see however the Realistic Storyline scenario in sect. 3.3 and the high SRM failure damage scenario in sect. 3.4).

Finally, in the Albedo Tipping Scenario (see sect. 3.4), we replace the damage tipping point described above by a tipping point which causes an additional radiative forcing (thought of as being due to temperature-driven albedo changes), loosely following Lemoine and Traeger (2014). The forcing obeys

$$F_{alb} = \alpha_{alb} \max(T - T_{alb}, 0) \tag{11}$$

i.e., a positive, temperature-dependent forcing occurs if the tipping point is activated and if $T$ exceeds the threshold $T_{alb} = 1.5K$. The tipping probability obeys eq. (10) except that the threshold $T_{tipp}$ is replaced by $T_{alb}$. Note that this tipping point is reversible in the sense that $F_{alb}$ can decrease again if $T$ decreases.

## 2.2  Optimisation and Performance Measures

As in DICE, we assume that all decisions are made by a single policy maker who aims to optimise the welfare of the (homoge-
neous) world population. As in DICE, welfare depends entirely on consumption. The economic output is spent on investment $I = rY$ and global consumption $Lc = Y - I$ where $r$ is the saving rate, and $L$ and $c$ are the world population and per capita consumption, respectively. We assume a fixed saving rate of $r = 22\%$. The utility $u$ (which can be thought of as the current "happiness" of the world population) depends on $c$: $u = L(c^{1-\gamma} - 1)/(1 - \gamma)$ with $\gamma = 2$ and the quantity to be maximised is the expectation value $\mathbb{E}(W)$ of the welfare $W$ (the time-integrated, discounted utility):

$$W = \sum_t u(t)e^{-\rho t} \tag{12}$$

where $t$ is (discrete) time and $\rho$ the rate of pure time preference. The greater $\rho$, the less does the far future count towards $W$. The morally correct value of $\rho$ has been fiercely discussed (Stern et al., 2007; Lilley, 2012; Ackerman, 2007). Here, we will not join the ethical debate on the 'correct' value, but use the standard value of 1.5% and perform a sensitivity study with $\rho = 0.5$ (see Sect. 3.5).

The decision variables are the amount of $CO_2$ abatement $\mu$ (the fraction of $CO_2$ avoided) and the sulphur injection rate $I_S$. The model is integrated in yearly time steps, but the decision variables $\mu$ and $I_S$ are changed only once a decade to save computational effort. The "policy" (sequence of values for $\mu$ and $I_S$) is optimised such as to maximise the expected welfare $\mathbb{E}(W)$ over a time horizon till 2400, though the far future is heavily discounted. The optimisation is performed using Dynamic Programming (see appendix). Once an optimal policy is found, it is evaluated by running an ensemble of 5000 members with
this policy and Monte-Carlo realisations of the stochastic elements (climate tipping and SRM failure). The best policy is the one yielding the highest expected welfare $\mathbb{E}(W)$ . For easier comparison, we define a performance measure based on the improvement of $W$ under policy $\pi$ with respect to a no-action policy ($\mu = 0, I_S = 0$):

$$\zeta(\pi) = 100\% \times \frac{W_\pi - W_0}{W_{AD} - W_0} \tag{13}$$

where $W_\pi$, $W_0$, and $W_{AD}$ are the expectation values over the Monte-Carlo ensemble of the welfare associated with policy $\pi$,
the no-action case, and the optimal policy for the deterministic Abatement-only case (the policy that would be optimal if the

decision maker may only use abatement, and no climate tipping occurs), respectively. By construction, the relative performance is 100% for Abatement-only in the deterministic case.

Although the objective for the optimisation is the expectation value of the welfare, it is also interesting to investigate the range of possible welfare outcomes, especially the worst (or at least relatively bad) case scenario. Hence we present two additional performance measures, based on the 10th and 90th percentile of the welfare $W_\pi$. Similar to eq. (13) we define $\zeta_X(\pi)$, the X-percentile relative performance of a policy $\pi$, as

$$\zeta_X(\pi) = 100\% \times \frac{W_{\pi,X} - W_0}{W_{AD} - W_0} \tag{14}$$

where $W_{\pi,X}$ is the Xth percentile of the welfare (discounted cumulated utility) for policy $\pi$. Note that $W_{AD}$ and $W_0$ are still the mean (i.e. not percentiles) welfare associated with the optimal policy for the deterministic Abatement-only case, and the no action case, respectively.

## 2.3 Scenarios

In Sect. 3.1 - 3.2 we first consider three stylised policy scenarios. The first is Abatement-only, in which the decision maker is allowed to use $CO_2$ abatement but no SRM. The second is SRM-only, in which the decision maker uses only SRM, until an SRM failure occurs, after which only abatement may be used. This scenario represents a society which does not reduce $CO_2$ emissions but relies entirely on SRM (until it fails). The third is Abatement+SRM, wherein the decision maker can use both Abatement and SRM, unless SRM fails, after which only abatement is used. A no-policy scenario with neither abatement nor SRM serves as benchmark for performance comparison (see eq. (14)). These three standard scenarios are first discussed in a deterministic setting (Sect. 3.1), i.e. in absence of climate tipping and SRM failure, before addressing them in the full model with uncertainty (Sect. 3.2).

While the previous stylised scenarios serve to isolate specific effects, we also present a more Realistic Storyline (see Sect. 3.3) which allows for the fact that it may take time to develop SRM technology, generate a legal framework and public support, and evaluate associated risks. Also, all these processes may fail or the effectiveness of SRM might be found too low. Therefore we assume that SRM will become possible only in 2055, and only at 30% probability. To be precise, at each time step until 2055, there is an equal probability that humanity discovers that SRM is impracticable. In the first decade where SRM is allowed, there is a 20% probability of SRM failure; in the second decade, 10%; in the third decade 5% and after that 1% per decade, i.e. after some decades of testing, failure becomes less likely. In this scenario, we also investigate the effect of a damage in case of SRM failure ('termination shock'): SRM failure is accompanied by a one-time reduction of the GDP by a factor $1 - \psi_{fail} I_S$ where $I_S$ is the injection rate in Mt(S)/yr and $\psi_{fail}$ is given in Table 1.

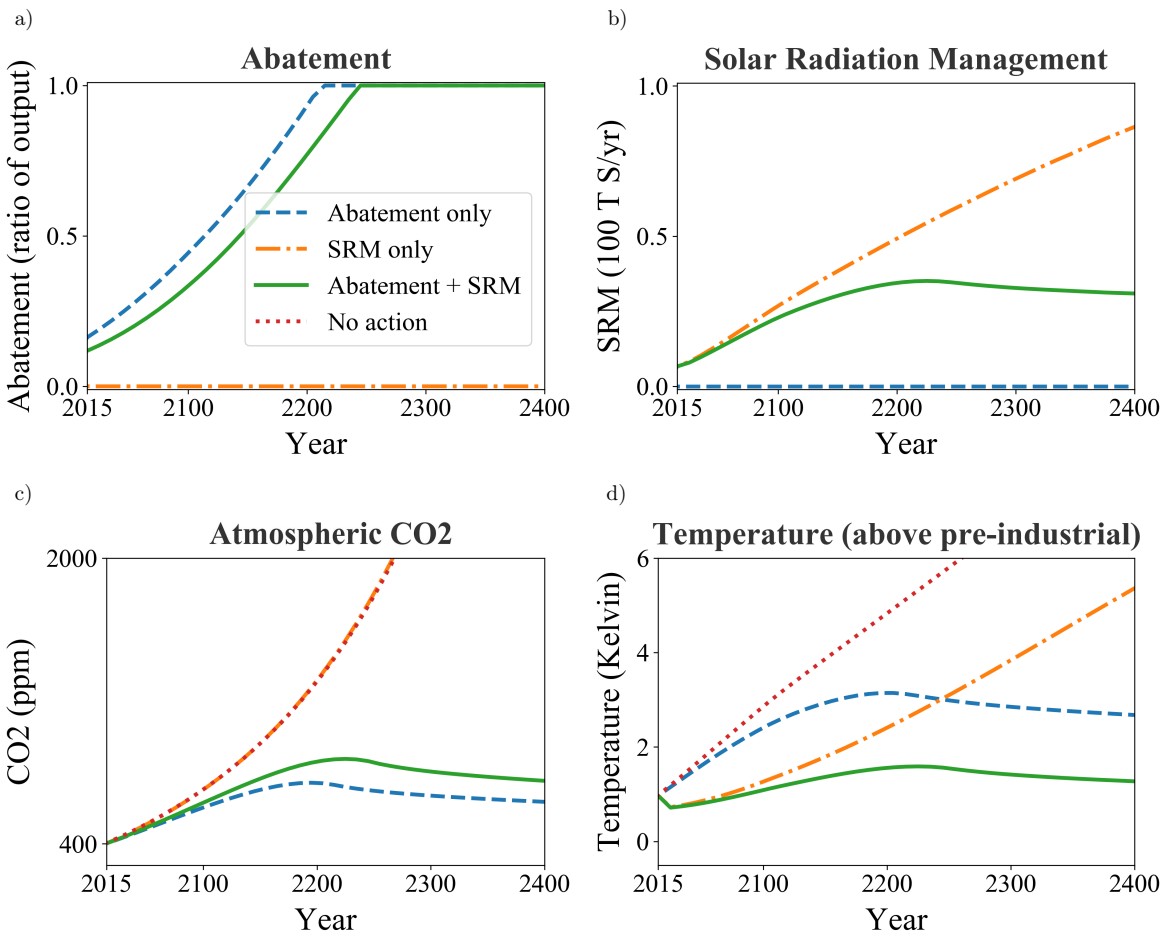

**Figure 2.** Optimal policy and climate model results for the three stylised scenarios in the deterministic setting. a) Abatement (fraction of $CO_2$ emissions avoided), b) SRM (Mt(S)/yr), c) atmospheric $CO_2$ concentration in ppmv, d) global mean temperature above pre-industrial (K). The blue dashed line represents the Abatement-only scenario, yellow dash-dotted: SRM-only, solid green: Abatement + SRM, red dotted (only plot c and d): no climate action (i.e. neither abatement nor SRM).

| Policy | $\zeta$ | peak SRM | Ab. 50% | Ab.90% | SCC |
|---|---|---|---|---|---|
| Abatement-only | 100% | / | 2114 | 2212 | 35 |
| SRM-only | 186% | * | / | / | 21 |
| Abatement+SRM | 238% | 35.1 | 2134 | 2243 | 20 |

\* SRM does not peak, but keeps increasing until the upper limit of $100 Mt(S)/yr$.

/  = Not applicable

**Table 2.** Comparison of policies in the deterministic setting (no tipping, no SRM failure). Abatement-only means that no SRM is used, SRM-only means that no abatement is used (unless SRM fails; see text), and in Abatement+SRM both are used. The performance $\zeta$ (see eq. (13)) is a measure of the increase in expected cumulated discounted utility w.r.t. the no-action scenario, and is normalised such as to yield 100% for Abatement-only. The column 'peak SRM' contains the highest SRM values (in Mt(S)/yr) over all time steps. 'Ab 50%' and 'Ab 99%' show the year in which the abatement reaches 50% and 99%, respectively. SCC is the social cost of carbon in the first time step (measured in \$(2005)/t(C)).

## 3  Results

### 3.1  The deterministic case

As a reference, we first consider the deterministic case, i.e. without SRM failure and tipping points, in the three stylised scenarios (see Fig. 2 and Table 2). Allowing SRM in addition to abatement delays abatement by 2-3 decades, but does not

replace it (Fig. 2a). $CO_2$ concentrations in Abatement+SRM (Fig. 2c) peak slightly later than in Abatement-only and reach higher values (875ppmv instead of 741ppmv). SRM helps to reduce global warming considerably: The global mean temperature change $T$ peaks at 1.6K for Abatement+SRM, but at 3.1K for Abatement-only (Fig. 2d). SRM slightly deceases towards the end of the simulation, when $CO_2$ concentration also goes down. This illustrates the potential use of SRM as a transition technology, especially under ambitious abatement: SRM can be used for a limited time in modest strength to cut off a warming overshoot.

In SRM-only, $CO_2$ concentrations reach 2000ppmv in 2260 and continue to increase (Fig. 2c). Note that currently known fossil fuel reserves are insufficient to generate this much carbon, but it is not impossible that fracking and newly discovered coal deposits will lead to sufficient fuel resources (Cassedy and Grossman , 2017). The temperature increase $T$ continues to rise, reaching 5.4K in 2400 (Fig. 2d), although it is lower than in the no-action case (neither SRM nor abatement). Due to the sub-linear increase of the radiative forcing with SRM, very SO$_2$ high injection rates would be needed to stabilise $T$ with SRM-

only, so that the damage related to sulphate injection outweighs the climate damages. Compared to SRM-only, considerably less SRM is needed in Abatement+SRM, namely $\approx 35 Mt(S)/year$ (Fig. 2b; $3-4$ Pinatubo eruptions per year), yet $T$ remains much lower. This suggests that abatement is required in order to achieve long-term temperature stabilisation.

The relative performance $\zeta(\pi)$ for SRM-only and Abatement+SRM, becomes 186% and 238%, respectively (see Table 2). (By construction, $\zeta = 100\%$ for Abatement-only in the deterministic setting.) The reason for the better performance of SRM-

only compared to Abatement-only is that SRM-only yields lower temperatures and a higher utility in the first two centuries,

| Policy | $\zeta$ | $\zeta_{10}$ | $\zeta_{90}$ | SRM fail | Tipping | peak SRM | Ab. 50% | Ab.90% | SCC |
|---|---|---|---|---|---|---|---|---|---|
| No action | 0% | | | / | 96.2% | / | / | / | 45 |
| Abatement-only (det. policy**) | 100% | | | / | 49.5% | / | 2114 | 2212 | 42 |
| Abatement-only | 105% | 77% | 121% | / | 37.8% | / | 2095 | 2215 | 41 |
| SRM-only | 181% | 179% | 185% | 19.8% | 60.96% | * | / | / | 23 |
| Abatement + SRM | 219% | 220% | 223% | 20.2% | 6.2% | 35.0 | 2139 | 2242 | 20 |
| Realistic Storyline | 125% | 78% | 190% | 79.9% | 30.1% | 31.4 | 2106 | 2234 | 37 |

* SRM does not peak, but keeps increasing until the upper limit of $100 Mt(S)/yr$.

** Tipping can occur, but the policy maker ignores this and chooses the policy which would be optimal in the deterministic case.

/ = Not applicable

**Table 3.** Comparison of policies in the stochastic setting, i.e. including climate tipping and SRM failure. No action means that neither abatement nor SRM are used; other scenarios are explained in Sect. 2.3. The perfomance measures $\zeta$, $\zeta_{10}$ and $\zeta_{90}$ are given in eq. (13) and eq. (14). The columns 'SRM fail' and 'Tipping' show the probability that SRM failure or climate tipping occurs before 2415. The column 'peak SRM' contains the highest SRM value (in Mt(S)/yr) over all time steps and over all ensemble members. This corresponds to members in which no SRM failure or climate tipping occurred, at least before the time of the SRM peak. 'Ab 50%' and 'Ab 99%' show the year in which the abatement reaches 50% and 99%, respectively. SCC is the social cost of carbon in the first time step (measured in $(2005)/t(C)).

which contribute most to the cumulative utility due to discounting. In addition, postponing damage is beneficial as it allows more time for accumulating capital.

## 3.2 The effect of uncertainties

Next, we include the stochastic elements, temperature-induced tipping and SRM failure, and determine again the optimal policies for each scenario, prior to evaluating the optimal policy by means of a Monte-Carlo ensemble (see Sect. 2.2). In Fig. 3b - l, we plot the policy (abatement and SRM) carbon concentration and temperature for the three stylised scenarios. The plots depict some sample paths of individual Monte-Carlo runs (thin blue lines), the range of possible outcomes (shading) and the ensemble mean (red line). For comparison, the results from the deterministic case (compare Fig. 2) are also plotted (blue dashed lines).

In the Abatement-only scenario, the danger of tipping initially leads to higher abatement (Fig. 3d) than in the deterministic case, although the temperature is not kept below the $2K$ threshold (see Fig. 3j). If tipping occurs, the abatement decreases again, as there is no further tipping point to be avoided. (This effect is caused by having a single tipping point which, once activated, does not react to system changes. Compare the Albedo tipping point in sect. 3.4.) The relative performance is 105% (see Table 3, row 3), i.e. it slightly improves when the decision maker takes tipping into account (compare Table 3, row 2). Recall that the reference scenario for $W_{AD}$ uses the policy that would be optimal in absence of tipping, i.e. the policy maker ignores climate tipping.

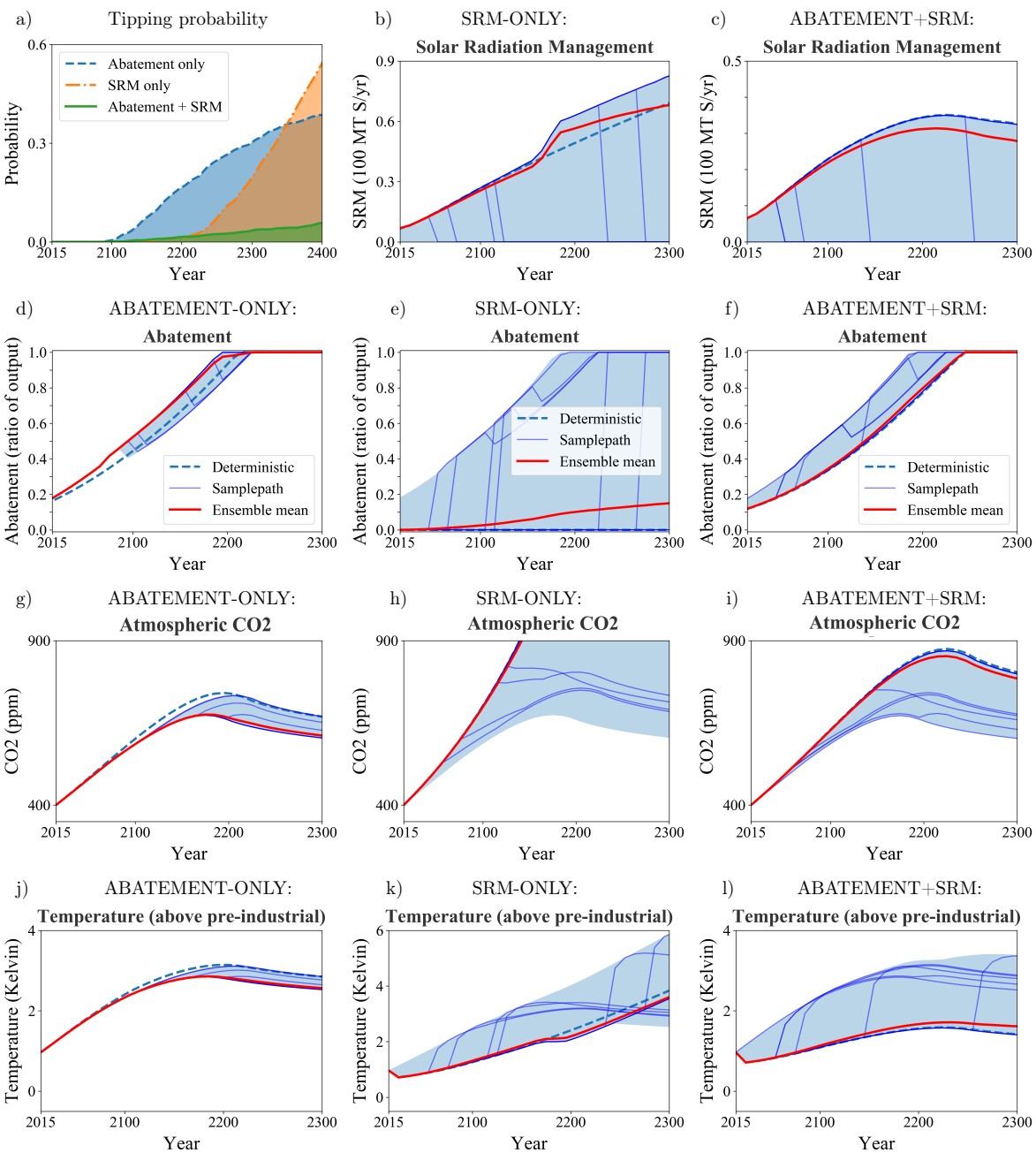

**Figure 3.** Tipping risk and policy in the stochastic setting (i.e. with tipping point and SRM failure). a) cumulative probability of tipping for Abatement-only (blue dashed line), SRM-only (yellow dash-dotted) and Abatement+SRM (green solid). b)-l): policy and climate response for the same scenarios (zoomed in on years 2015-2300 to enhance readability), namely Abatement-only (left column, plots d),g),j)), SRM-only (middle column, plots b),e),h),k)), and Abatement+SRM (right column, plots c),f),i),l)). Variables shown are: SRM deployed (first row, plots b), c)); note the different y-axis scale), abatement fraction (second row, plots d),e),f)), atmospheric CO2 content in ppmv (third row, plots g),h),i)), and global mean temperature change (last row, plots j),k),l); note the different y-axis scale). The thin blue lines represent a sample of individual ensemble members, the thick red line the ensemble mean, and the blue shaded area indicates the range of possible values in the whole ensemble. The dashed blue line depicts the results from the deterministic case (Fig. 2) for reference.

In the Abatement+SRM case, the optimal policy closely resembles the deterministic one if no SRM failure occurs (Fig. 3c,f). Without SRM failure, $T$ stays below $2K$ (Fig. 3l), and hence no tipping occurs. In case of an SRM failure, the temperature suddenly increases and abatement suddenly increases, as the decision maker now tries to limit the warming (and tipping risk) with only abatement. Note that such a sudden increase in abatement may not be feasible in reality. If climate tipping occurs,
abatement is reduced again. Compared to Abatement-only, the abatement is delayed by 3-4 decades.

In the SRM-only scenario, the policy again resembles the deterministic case provided no SRM failure occurs and $T$ is below 2K (Fig. 3b). When $T = 2K$ is reached, SRM increases sharply to reduce the tipping risk. As before, abatement strongly increases after SRM failure, but is reduced slightly if tipping occurs (Fig. 3e). SRM-only has a performance of 181%, much higher than Abatement-only. However, the chance of climate tipping by the year 2415 is considerably higher for SRM-only
(61.0% vs 37.8% for Abatement-only, see Table 3). As in the deterministic setting, the reason is that initially SRM can control the global warming more effectively than abatement, while abatement is a long-term measure. Hence damage is postponed to the far future which is heavily discounted. The cumulative probability of tipping is lower for SRM-only than for Abatement-only until 2350, when the situation reverses (Fig. 3a).

Compared to the deterministic cases, including uncertainty slightly reduces the difference in relative performance between
Abatement-only and the scenarios using SRM (compare Table 3 vs Table 2). There are two competing effects: The danger of tipping might favour using SRM, which reduces the tipping probability in the near future, while the possibility of SRM failure reduces the performance of SRM-based scenarios.

In Abatement-only, there is a high spread between the relative performance measures $\zeta$, $\zeta_{10}$ and $\zeta_{90}$, compared to SRM-only and Abatement+SRM. This is due to the fact that in most ($> 90\%$) of the ensemble members, SRM keeps global warming below
2K at least until $\approx 2200$. Hence SRM postpones climate tipping into the far future (except in the few ensemble members with early SRM failure), while for Abatement-only, tipping can occur as early as 2080. Early tipping greatly reduces the performance because it reduces the GDP for a long period of time and because it is less heavily discounted. For Abatement+SRM, only 6.2% (i.e. $< 10\%$) of the ensemble members show climate tipping, but they strongly affect the mean performance. This explains why, for this scenario, $\zeta < \zeta_{10}$.

Although DICE is too limited to give reliable absolute values of the Social Cost of Carbon (SCC) (van den Bergh and Botzen, 2015), comparing scenarios gives qualitative insight into how SRM affects the SCC (Table 2 and Table 3). For Abatement-only, the SCC in 2015 is $35\$/t(C)$ (in 2005\$) in the deterministic case and $41\$/t(C)$ when including tipping points. For Abatement+SRM, the SCC is $20\$/t(C)$ (both deterministic and stochastic): SRM lowers the SCC by partially compensating the damage caused by $CO_2$ emissions. For SRM-only, the SCC is only slightly higher, namely $21\$/t(C)$ (deterministic) and
$23\$/t(C)$ (stochastic), because SRM suppresses most climate damage in the near future, which is discounted least.

### 3.3 Realistic Storyline

The previous scenarios were very stylised, in order to isolate the impact of SRM and stochastic elements. However, the actual situation is more complex: Presently SRM is not available and we do not know whether it ever will be; yet we might want to decide now whether to pursue (research and development of) SRM. To address this question, we consider a "Realistic Storyline"

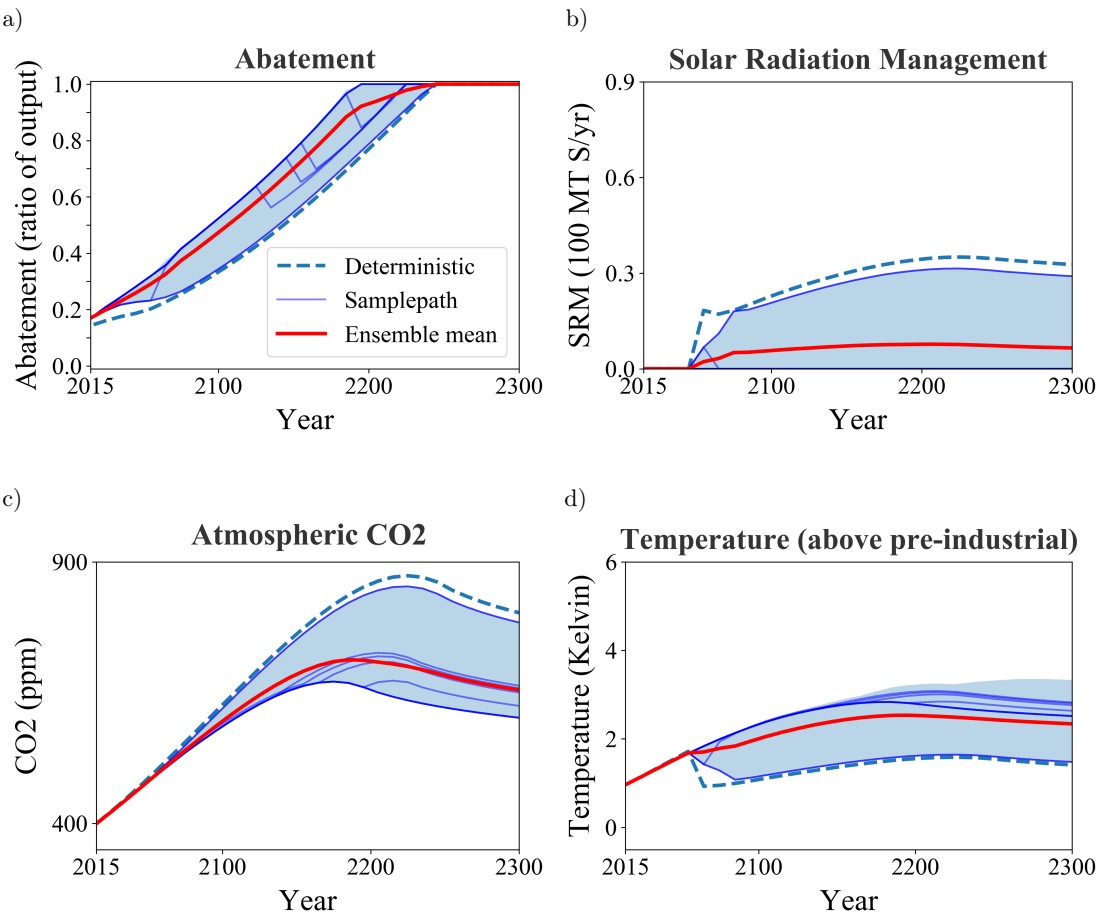

**Figure 4.** Optimal policy and climate development for the Realistic Storyline scenario. a) Abatement fraction, b) SRM in 100 Mt(S)/yr, c) atmospheric carbon concentration in ppmv, d) global mean temperature change w.r.t pre-industrial. The thin blue lines represent a sample of individual ensemble members, the thick red line the ensemble mean, and the blue shaded area indicates the range of possible values in the whole ensemble. The dashed blue line depicts the results from a deterministic reference case in which SRM becomes available in 2055 certainly and neither SRM failure nor climate tipping occur.

scenario, in which we assume that SRM will become possible only in 2055, and only at 30% probability (in the decades before 2055, there is a certain probability each time step that SRM is declared infeasible, e.g. because scientists identify unacceptable environmental risks). We also assume that after 2055, the probability of SRM failure decreases in time, i.e. with ongoing testing, and allow for damage associated with a termination shock in case of SRM failure (see Sect. 2.3). Unlike (irreversible)
climate tipping, the termination shock is a short-lived phenomenon and is stronger for large SRM.

In those ensemble members where SRM becomes available in 2055, it is used sparingly in the first time step, because the probability of failure is still high and the decision maker wants to limit the termination shock. In later time steps, SRM is used only slightly less than in the Abatement+SRM scenario, peaking at 31.4% rather than 35%. This difference mainly arises because the decision maker wants to reduce the termination shock: If the termination shock damage is omitted from the
Realistic Storyline, SRM peaks at 34.7%.

In the first time step (2015), when the decision maker assumes that SRM will become available with 30% probability only, the abatement is $\mu(2015) = 0.17$, only slightly less than in the Abatement-only scenario where $\mu(2015) = 0.18$. For comparison, in a deterministic reference case in which SRM will be available from 2055 certainly, and no SRM failure or tipping occurs, $\mu(2015) = 0.14$ (see Fig. 4). As time progresses until 2055, the ensemble members diverge: If SRM is already
banned, abatement increases, but if a time step has passed without a ban, the decision maker becomes more optimistic that SRM will become feasible and abatement becomes less ambitious. In ensemble members where SRM becomes available, 50% abatement is reached 45 years later than in cases where SRM remains impossible. For current policy, however, the most important point is that in 2015 ("now"), the 30% chance of SRM becoming available does not lead to significant reduction in optimal abatement.

On the other hand, the performance $\zeta$ of this scenario is 125% (Table 3), significantly higher than for abatement-only. The lowest 10-th percentile performance, $\zeta_{10}$ is very similar to the Abatement-only scenario. In the Realistic Storyline, the low-performance members are those in which SRM never becomes available, and the policy (i.e. trajectory of abatement) in these runs is very similar to Abatement-only. However, $\zeta_{90}$ is much higher for the Realistic Storyline than for Abatement-only. This measure is dominated by those members in which SRM becomes available. The total climate tipping risk for the Realistic
Storyline is 30.1%, compared to 37.8% in the Abatement-only scenario. The SCC for the Realistic Storyline is $37\$/t(C)$, 12% lower than for Abatement-only.

These comparisons between the Realistic Storyline and Abatement-only indicate that the former performs better. This is because in those cases where SRM does become available, the welfare gain of climate policy is twice as high as in the Abatement-only case. Therefore, a policy maker in 2015 should not dismiss SRM prematurely, but keep the option open (by encouraging
research and development). If we are lucky and SRM works well, it can greatly enhance future welfare, whereas if it never becomes feasible, we are not worse off than with abatement-only. (Note, however, that we did not include the possibility of a large-scale SRM test with huge unexpected damage, but assumed careful, well-designed research.) However, the prospect of possible future SRM should not lead to a significant reduction in abatement efforts at the current stage.

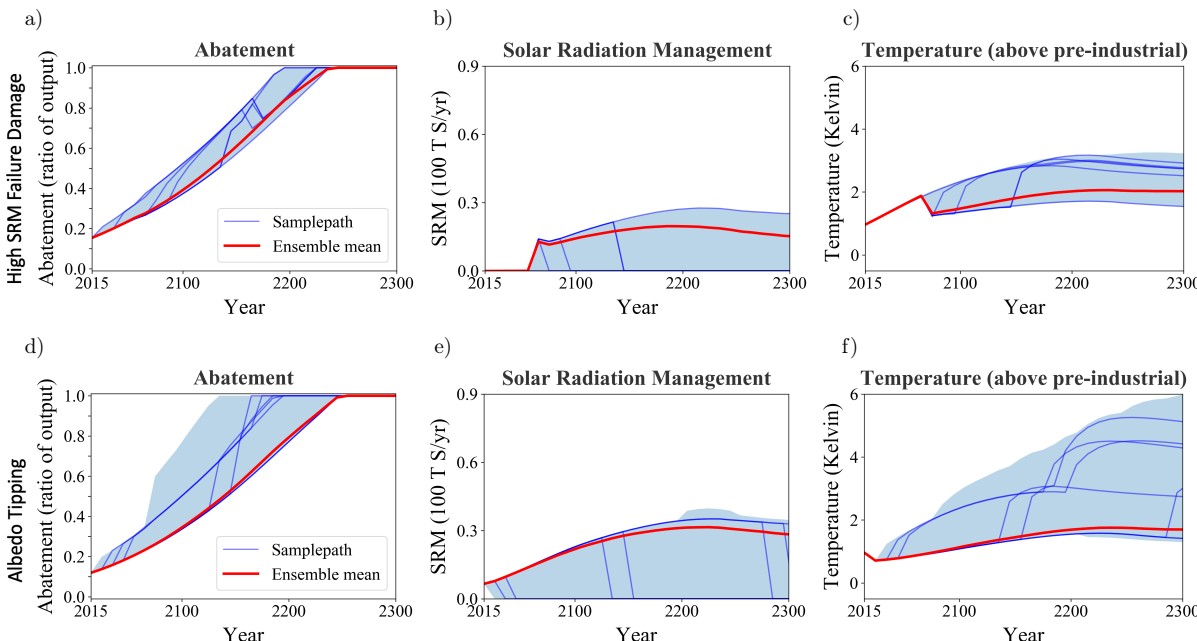

**Figure 5.** Policy and temperature for the 'SRM as insurance' scenarios (see sect. 3.4). The top row shows the scenario with high damage in case of SRM failure, while the bottom row shows the scenario with an albedo tipping point. The left column (a,d) show abatement, the middle one (b,e) SRM, and the right (c,f) warming. The thin blue lines represent a sample of individual ensemble members, the thick red line the ensemble mean, and the blue shaded area indicates the range of possible values in the whole ensemble.

## 3.4 SRM as 'Climate Insurance'?

In the previous scenarios, SRM was used in a continuous way as a complement for abatement in order to further reduce global warming, especially when the warming was highest. Here we investigate under which circumstances it can be advisable to use SRM as an 'insurance', that is, suddenly increase its use or even voluntarily delay using it at all.

First, we consider a situation in which SRM is very dangerous, and thus unattractive to use unless climate change is also very dangerous. This is achieved by assigning a very high, but one-time, damage to SRM failure, namely reducing capital by a factor $\Omega_K = I_S/(I_S + I_{S0})$ in case of SRM failure. Here $I_S$ is the injection rate in $Mt(S)/yr$ and $I_{S0} = 5Mt(S)/yr$. This means that already at modest injection rates, SRM failure is assumed to cause substantial capital losses. In addition, we increase the likelihood of tipping failure by a factor of 4. Apart from these changes, the scenario is the same as the Abatement+SRM scenario in sect. 3.2. This scenario is not necessarily considered the most likely, but serves as a proof of concept. The result is that SRM is not until the tipping threshold $T = 2K$, threatens to be reached (see fig. 5a-c). When the threshold is reached, SRM is started and somewhat more SRM is applied than strictly necessary to keep below $T + 2K$. This is because in our parametrisation, damage levels off somewhat with increasing injection rate, i.e. if SRM is used at all, then a little extra doesn't make failure costs that much worse. The temperature is kept below $T = 2K$ throughout, unless SRM fails. Compared to the

standard Abatement+SRM case, peak SRM is reduced to $27.6 Mt(S)/yr$, i.e. by about 21%, and 50% abatement is reached in 2127, i.e. 12 years earlier. This experiment shows that the possibility of SRM causing high damage can cause a delay in its use until climate change also becomes very dangerous (tipping threshold reached).

Second, we replace the standard tipping point in Abatement+SRM by the 'albedo' tipping point (see sect. 2.1.4). It is found
that if the policy maker can use SRM freely, he does not employ it to such a degree as to stay below $T = T_{alb} = 1.5K$, but takes the (small) chance of crossing the threshold. If this happens, he does increase SRM to counteract the albedo feedback (the 'bump' after 2200 in fig. 5e)). Although the time step for determining policies is 10 years, the albedo feedback is weak enough that no runaway global warming occurs, since with SRM, $T - T_{alb}$ and hence $F_{alb}$ is small. This is why a modest increase suffices to suppress this effect. However, if SRM has failed, temperature is much higher than $T_{alb}$, increasing both
the probability of albedo tipping and the radiative forcing strength if tipping occurs. As in the standard 'Abatement+SRM' scenario, the policy maker increases abatement in case of SRM failure to avoid the tipping point. However, if the albedo tipping occurs after SRM failure, the policy maker increases abatement yet again in order to limit the positive temperature feedback. Nonetheless, the albedo tipping can cause additional warming of more than 2K. A positive climate feedback tipping point can thus lead to enhanced climate policy - SRM or abatement or both - after being triggered, in order to reduce its
consequences.

### 3.5  Sensitivity Analysis

The results of our model substantially depend on the rate of time preference $\rho$ (see Sect. 2.2). In Abatement-only, reducing $\rho$ from the standard value of $1.5\%$ to a lower value of $0.5\%$ will lead to stronger abatement: 50% abatement is reached 27 years earlier (see Table 4). This is expected, as a lower rate of time preference means that the decision maker gives more weight
to the welfare of future generations and is more willing to sacrifice present consumption to reduce climate change. The SCC rises from $\$41/t(C)$ to $\$70/t(C)$. Interestingly, abatement also increases in the Abatement+SRM scenario (50% abatement is reached 23 years earlier) when reducing $\rho$ to $0.5\%$, while the peak SRM (definition: see Table 3) decreases by about 11%. In other words, a decision maker who cares strongly about the future will choose to reduce $CO_2$ emissions rather than forcing future generations to rely on SRM, which causes damages and might fail. The SCC rises from $\$20/t(C)$ to $\$30/t(C)$.
A potentially important limitation of DICE is that abatement costs are exogenous, whereas in reality one would expect costs to decline with growing employment (learning-by-doing). While fully exploring learning-by-doing is outside the scope of this study, we estimate the sensitivity to abatement costs in a simulation wherein abatement costs decrease more quickly in time and reach a lower value for $t \to \infty$. This is done by putting $\lambda_2 = 1.5, \lambda_3 = 0.015$ in eq. (8), which lowers abatement cost by a factor of about 0.6 after 70 years, compared to the standard scenario. The resulting policy shows a faster abatement by about
30 years, leading to a lower peak in atmospheric carbon (745ppm instead of 870ppm). Peak SRM is reduced to 29Mt(S)/yr, as less SRM is needed if carbon concentrations are lower. Thus the development of abatement cost can significantly affect the need for SRM.

The distribution of the damages between the two major contributors, namely warming and residual climate change, was chosen rather arbitrarily. However, halving $\psi_T$ (warming contribution) and doubling $\psi_P$ (residual contribution) does not

| Scenario | Abate 50% | peak SRM | SCC |
|---|---|---|---|
| Abatement-only, standard | 2095 | / | 41 |
| Ab.+SRM, standard | 2139 | 35.0 | 20 |
| Abatement-only, low rate of pure time preference ($\rho = 0.5\%$) | 2068 | / | 70 |
| Ab.+SRM, low rate of pure time preference ($\rho = 0.5\%$) | 2116 | 31.1 | 30 |
| Faster decline abatement cost ($\lambda_2 \to 2; \lambda_3 \to 0.015$) | 2112 | 29.0 | 21 |
| Ab.+SRM, less temp. damage, more precip.damage ($\psi_T \to \psi_T/2, \psi_P \to \psi_P \times 2$) | 2143 | 32.6 | 17 |
| Ab.+SRM, lower tipping threshold ($T_{tipp} = 2K \to 1K$) | 2139 | 35.6 | 21 |
| Ab.+SRM, double damage from tipping ($\Omega = 0.8$) | 2136 | 34.8 | 20 |
| Ab.+SRM, double climate tipping probability ($\kappa_{tipp} \to \kappa_{tipp} \times 2$) | 2137 | 34.9 | 20 |
| Ab.+SRM, quadrupled SRM failure probability ($\kappa_{fail} \to \kappa_{fail} \times 4$) | 2121 | 34.3 | 23 |
| Ab.+SRM, double damage from SRM ($\psi_S \to \psi_S \times 2$) | 2133 | 26.8 | 22 |
| Ab.+SRM, half damage from SRM ($\psi_S \to \psi_S/2$) | 2143 | 43.6 | 20 |

**Table 4.** Policy metrics of the sensitivity runs. 'Abate 50%' is the year in which Abatement reaches 50% ($\mu = 0.5$). 'peak SRM' (in $Mt(S)/yr$) is the highest SRM value of the ensemble (over all times and all members) and corresponds to those ensemble members without early SRM failure or climate tipping. 'SCC' is the social cost of carbon in $\$(2005)/t(C)$. All simulations were preformed in the stochastic settings and are either Abatement-only or Abatement+SRM (abbreviated here as Ab.+SRM). The first two cases, labelled 'standard', are repeated from Table 3 for convenience. The sensitivity runs correspond to those discussed in Sect. 3.5.

qualitatively affect our results. 50% abatement is reached 4 years later in the Abatement+SRM scenario, and SRM peaks at 32.6Mt(S)/year instead of 35.0Mt(S)/year, i.e. the optimal policy still combines a similar abatement with modest SRM. The SCC drops from $\$20/t(C)$ to $\$17/t(C)$. Lowering the tipping threshold from $2K$ to $1K$ leads to a 2% increase in peak SRM, while not affecting abatement. Doubling the damages associated with climate tipping ($\Omega = 0.9 \to \Omega = 0.8$) only accelerates

5    50% abatement by 3 years in the Abatement+SRM case and the SCC remains at $\$20/t(C)$. Doubling the likelihood of climate tipping ($\kappa_{tipp}$) accelerates 50% abatement by 2 years and likewise does not affect SCC. Increasing the failure probability of SRM ($\kappa_{fail}$) by a factor of 4, i.e. such that SRM failure occurs in 80% of the ensemble members rather than 20%, increases the SCC only by 15% in the Abatement+SRM scenario, i.e. from $\$20/t(C)$ to $\$23/t(C)$.The reason is that the likelihood of SRM failure in the first decades, which are least discounted, is still fairly small. The peak SRM is reduced only by 2%: As

10   long as SRM is available, it is used despite high failure probability. 50% abatement is reached in 2121, rather than 2138, in the ensemble mean. Doubling the damage associated with SRM (i.e. doubling $\psi_S$) accelerates 50% abatement by 6 years and the SCC rises from $\$20/t(C)$ to $\$22/t(C)$. The peak SRM is reduced by about 23%, to 26.8Mt(S)/yr. Likewise, halving $\psi_S$ increases peak abatement by 25%. Hence even if SRM is twice (or half) as damaging as assumed in the standard case, the optimal policy still employs modest SRM as complement to abatement.

To summarise, changes in the damage function and/or likelihood of stochastic events do not qualitatively affect the optimal policy in the Abatement+SRM scenario, which consists of a combination of reasonably high abatement (delayed by a few decades w.r.t Abatement-only in the standard settings) and modest SRM.

## 4  Summary and Discussion

In this paper, we present the first cost-benefit analysis of SRM under uncertainty performed with a rigorous optimisation approach (dynamic programming). From our analysis we draw two conclusions. First, sulphate SRM has the potential to greatly enhance future welfare and should therefore be taken seriously as possible policy option. Second, even if successful, SRM does not replace $CO_2$ abatement, but complements it. In particular, a policy maker who puts great value on the welfare of future generations (i.e. uses a low rate of pure time preference) will accelerate abatement efforts, which have a long-term

benefit, rather than forcing later generations to rely on SRM. Apart from smoothly reducing peak warming, SRM might also have a role to play as emergency measure, e.g. in case of emerging positive warming feedbacks or unforeseen strong climate-induced damages. However, this might be a risky approach if SRM itself is potentially associated with strong damages.

Compared to previous studies (Goes et al., 2011; Moreno-Cruz and Keith, 2013; Heutel et al., 2018), our results are more optimistic about SRM, which seems partly due to the improved methodology we adopted. For instance, demonstrating that

welfare is severely impacted if the decision maker makes wrong assumptions on the SRM-related damages (Bahn et al., 2015) is not a consistent cost-benefit analysis. The analysis by Goes et al. (2011) only considers a full replacement of abatement by SRM, rather than a complementary approach. Compared to Heutel et al. (2018), we find a much stronger reduction in the SCC. However, as discussed previously, their model and optimisation method differ in some crucial points from ours. In particular, Heutel et al. (2018) assume that the implementation cost and damage associated with SRM depend on the fraction

of CO2-induced radiative forcing that is balanced by SRM - no matter how high the CO2 concentration is - rather than letting costs and damage depend on the amount of sulphur injected. Therefore at high (low) CO2 concentrations, they obtain a much higher (lower) radiative forcing effect from SRM for the same price, which makes SRM more (less) attractive. In their deterministic simulation, they compensate 50% of the peak CO2-induced radiative forcing of $6W/m^2$, which in our model settings would require injection rate of $27Mt(S)/yr$ - about 80% of our peak injection rate of $35Mt(S)/yr$ in the

deterministic Abatement+SRM scenario. However, in the first century, Heutel et al. (2018) use considerably less SRM because they overestimate the price by ignoring that much lower injection rates are needed while CO2 concentrations are low. So overall they use too little SRM and therefore end up with higher temperatures (about $2.5K$ peak warming) and a higher SCC.

Our results should not be interpreted as precise policy recommendations to set, say, exact values of the SCC, as our model is too limited to offer more than a qualitative exploration and comparison of simple scenarios. For example, uncertainty in the

climate system is limited to one tipping point, while uncertainty in the climate sensitivity is ignored. Our climate model is based on linear response theory, and although this approach captures many climate feedbacks adequately, it does not capture the possible dependence of the response on the background state, e.g. a saturation of carbon sinks (Aengenheyster et al., 2018).

A controversial component in Integrated Assessment Models such as DICE is the quantification of climate damages (Howard, 2014; van den Bergh and Botzen, 2015), which is highly aggregated and based on very limited data. We introduced additional parameters to the damage function by making a plausible, but rather ad-hoc attribution of climate damages to temperature, global precipitation ('residual climate change') and $CO_2$ concentrations. Also little is known about the size of ecological, let alone economic, damages associated with SRM. Gaining a better understanding of these damages, and those related to climate change, is essential for conducting a meaningful cost-benefit analysis and ultimately determining a climate policy, hence it should be given a high priority.

The abatement sector of DICE also has important limitations. First, technological improvement is exogenous (abatement costs decrease in time at a prescribed rate), rather than including learning-by-doing (costs decrease with technology employment). This means that in DICE it is advantageous to wait for the later cost reduction, rather than starting early to bring abatement price down through learning. In addition, DICE assumes that abatement is always costly, whereas in fact, the energy transition might rather be a big investment: Once the infrastructure is installed, green energy might be cost-competitive with fossil fuels. Both effects likely bias our results against early abatement. A faster (still exogenous) decrease in abatement costs was found to lead to faster abatement and reduced peak SRM. Our model does not include negative emission techniques, which might provide an important alternative to SRM. Neither does it include active adaptation. The trade-off between negative emissions, adaptation and SRM would be interesting to study with a more detailed model. Finally, DICE assumes a homogenous economy and a single decision maker. In reality, the damages and benefits of SRM are likely unevenly distributed, with potential for solitary actions and conflict, which was not studied here.

Despite the large scientific and political uncertainties which need to be overcome, we believe that one cannot afford to dismiss SRM at the current stage, as it has the potential to greatly reduce climate risk and enhance future welfare. However, the scientific uncertainties, especially concerning efficiency and damages of SRM, as well as the extent by which SRM can mitigate damages inflicted by global warming, must be better quantified. For the time being, the uncertain prospect of SRM becoming available should not tempt us to reduce abatement.

*Code availability.*

*Data availability.* The code used (described in the Methods section) is available upon request from the corresponding author.

*Code and data availability.*

*Sample availability.*

**Appendix A:  Solving the GeoDICE model**

**A1    Terminal function**

Unrealistic behaviour occurs in the last time steps of an optimisation problem, because the decisions made do not influence
the future anymore (as the future is not simulated). To avoid this problem, we follow Cai et al. (2016) and run the optimisation
over 600 years, while only considering the first 400 as actual simulation and the final 200 years as 'terminal function'. During
termination, tipping can still occur and SRM can be freely chosen, while abatement is set to 1. Due to discounting, the trajectory
after 600 years has little relevance for the optimal policy during the first 400 years. Indeed, prolonging the runs to 800 years
had a negligible effect on policies during the first 400 years.

**A2    Optimisation method**

The social planner problem aims at finding the policy that maximises the expected cumulative discounted utility. To solve this
problem in the stochastic setting, we apply dynamic programming (Bellman, 1957). This methodology relies on the concept of
the value function to obtain the optimal policy via backward reduction. As our state space is continuous and no analytic solution
is available, we are forced to adopt some approximation scheme to represent the value function at each time step. Following
Cai et al. (2016), we use a Chebyshev approximation, which is well suited for parallelisation. The Chebyshev polynomial is
obtained by solving a small optimisation problem at each of a finite number of regularly spaced Chebyshev approximation
nodes. We used a fourth-degree Chebyshev polynomial with five approximation nodes per continuous dimension. In combina-
tion with the binary state variables for the tipping point and SRM failure, this results in 312500 approximation nodes per time
step. This method is developed and discussed extensively in the work by Cai (2009); Cai et al. (2012a, 2016). For a complete
overview we refer the reader to these papers and the references therein. Here we outline the methodological choices specific to
the present application: the boundaries used for the domain of the Chebyshev polynomial, and adjustments to the value function
approximation to accommodate the asymmetry and non-smoothness of the true value function. Additionally, we examine the
accuracy of this methodology when applied in the current setting.

**A2.1    Boundaries**

In order to define the Chebyshev approximation nodes, we must first set the boundaries of the region of state space in which
we are interested. To do this, we calculate three trajectories in the deterministic model: first, the optimal trajectory (obtained
by optimising the whole system in all decision variables with standard deterministic optimisation software); second, a 'high-
emission' trajectory calculated by setting mitigation and SRM to zero for the whole run; and third, a 'low-emission' trajectory,
calculated by setting mitigation to one and SRM to zero for the whole run. We subsequently take as domain boundaries for each
variable the minimum and maximum over these three trajectories, with an additional margin of minus, respectively plus 30%

of these values. For all experiments, we check that all the sample paths in the ensemble remained well within the boundaries of the domain. For approximation nodes close to the boundaries, it will still be possible to select actions that may bring the system outside the boundaries in the next step. Since a Chebyshev polynomial cannot be extrapolated outside its domain, we first project the state onto the region of interest before evaluating the approximate value function.

## A2.2 Value Function Smoothing

In the current setting, directly using a Chebyshev polynomial to approximate the value function gives poor results because the value function exhibits an asymmetry and a non-smoothness that a low-degree Chebyshev polynomial cannot capture. The discontinuity is caused by the fact that in states with positive temperatures, SRM is available to reduce them, while in states with negative temperatures this is impossible: therefore, positive temperature deviations are preferred over negative temperature deviations of equal magnitude. This problem is resolved by allowing 'reverse' SRM, which generates a radiative forcing of the same magnitude but opposite sign as regular SRM. Allowing such actions changes the value of certain states, thus removing the asymmetry. This is a purely mathematical construct (we do not assume such reverse SRM is actually possible): the states with modified values are never reached in actual trajectories, and are only considered in the first place because the domain of the Chebyshev approximation must be a hypercube.

The non-smoothness results from the fact that the tipping point can only be crossed after a certain threshold is reached: this generates a discontinuity in the first derivative of the value function. This is resolved by fitting two separate Chebyshev polynomials to the two parts of the value function.

## A2.3 Accuracy

We test the accuracy of our optimisation by comparing the resulting policy in a deterministic setting to the policy obtained by regular non-linear optimisation. The difference in action and trajectory is < 3%, while the difference in the SCC is < 2%. For the scenario in which only abatement is allowed, errors are lower (0.1−1% for actions and SCC, 0.01−0.1% for trajectories), which is in line with the accuracy reported by Cai et al (Cai et al., 2016). Good accuracy in the deterministic setting may not generalise to the stochastic setting when the stochasticity itself introduces issues. To guard against this problem we ensure that the value function approximation fits well to the actual value function samples obtained at each time step.

*Author contributions.* All authors conceived research. C.E.W. and K.G.H. designed model and scenarios and interpreted results. K.G.H. performed simulations. All authors contributed to writing the paper.

*Competing interests.* The authors declare no conflict of interest.

*Disclaimer.*

*Acknowledgements.* Claudia E. Wieners is supported by the Complexity Lab Utrecht (CLUe) of the Centre for Complex Systems Studies at Utrecht University. Henk A. Dijkstra acknowledges support by the Netherlands Earth System Science Centre (NESSC), financially supported by the Ministry of Education, Culture and Science (OCW), Grant no. 024.002.001.

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
