# Peer review of "Complementing CO2 emission reduction by Solar Radiation Management might strongly enhance future welfare"

_Earth System Dynamics, 2019_

## Referee Comment (RC1) · Michael MacCracken (Referee) · 23 Feb 2019

Overall Recommendation: The basic work is interesting and potentially publishable, but in my opinion, there are serious deficiencies in the degree of context included in the text that simply have to be included in a revision of the paper.

General Comments

1. The authors are commended for taking on the task of seeking to do a comparative analysis of the relative merits of a wider set of policy options for responding to human-induced climate change than is traditionally done. This is both appropriate and needed.

[Figure]

While I have quite a number of comments about the paper and what more I think is needed, this is not to say that what has been done is a useful advance and I would encourage the authors to keep at this, recognizing that what has been achieved is a starting point and that what will ultimately be needed is a good bit more.

2. Regarding the overall thrust of the article, use of the term "Geoengineering" in the title conveys the impression, at least to me, that both Carbon Dioxide Removal (CDR) and Solar Radiation Management (SRM) will be considered. The text actually speaks almost exclusively about Solar Radiation Management, and this is really unfortunate because CDR is really the potential exit strategy for SRM rather than, as in this article, assuming SRM would go on indefinitely. So, one thing missing at the front of the article is a bit of discussion about the various options for responding, which include reducing emissions (normally called mitigation, here called abatement), adaptation, CDR and SRM. And the mitigation or abatement can involve both cutting emissions of long-lived species like CO2 and short-lived species like methane, tropospheric ozone, and black carbon. This article thus only focuses on CO2 mitigation and SRM, and this needs to be more clearly indicated. One could perhaps say that mitigation includes CDR, but then one would also need to be allowing negative net emissions, which is actually something that the IPCC's recent report ends up on relying on.

3. It is really not made nearly clear enough that what the DICE model calculates is not related to actual climate change impacts, but a "damage function" that is the projected change in the size of GDP. So, while I was expecting the paper to be considering such aspects as impacts on agriculture, forests, ecosystems, coral reefs, biodiversity loss, etc.; the impacts due to sea level rise, extreme weather events, wildfires, etc. The problem with this is that while it is interesting to have an indication of the overall effect on the economy, it is not possible to distinguish the changing share of the economy going to improving general welfare and the share going to recover from disasters, the need to rebuild, the relocation of refugees, and so on. While those familiar with DICE might be aware of this, this paper really needs to be clear on all of this; I just think using

the phrase "damage function" really needs to be clarified. I'd note then that with this construction of the issue, it becomes hard to do more than quite simply represent the tipping point, etc.

4. While the result and conclusion seem plausible, in reading the paper I kept thinking about the shortcomings and limitations of what was being done, even when considering only mitigation/abatement and SRM. Overall, I found it frustrating to only get an overview of the limitations at the end of Section 4 of the article rather than being made aware of the limitations up front and having the limitations then related to the various statements along the way. While I think this paper is an advance, not really clearly indicating the limitations up front and providing some explanation of their potential significance will, I think ,frustrate readers, whereas having the limitation up front and then in Section 2 indicating all that has to be done to get to the advanced point that is reached will better allow appreciation of the challenge posed by this type of analysis, necessary as it is.

5. Another general issue that bothered me was that conclusions were offered on various aspects of the issue (e.g., uncertainties being large) without giving a comparison to anything. Basically, the question posed is whether the world would be better off going through greenhouse gas-induced climate change with or without invoking SRM, so it seems to me that every time some conclusion is proposed about something like uncertainties, the context has to be the other possible option. So, are uncertainties of SRM really large compared to uncertainties of uncontrolled warming (so both with and without abatement), for example. Are uncertainties about having the climate more or less as at present as a result of the equivalent of a human-induced volcanic eruption larger or smaller than the uncertainties of the impacts and results of the climate heading toward a 3-4 C warming (uncertainties considered small enough to justify an international decision to phase out the global fossil fuel energy system later this century)? By not summarizing up front the uncertainties present with respect to projections of global warming of a few degrees C (among which is the issue of tipping points—and there

are likely many, and some are irreversible, like biodiversity loss; some, like the commitment to melting of ice sheets, have long lag times; etc.), there is really no context for making a statement about whether the uncertainties of SRM are important or not. It seems to me that this just has to be provided.

Specific Comments

Page 1, Title: Given the general comment, the word "Geoengineering" in the title needs, in my view to be reconsidered. Perhaps use "Climate Intervention" or something similar so it is clear that CDR is not being considered.

Page 1, line 2: Change to "to lower". Also, the word "may" seems totally inadequate. First, use of this word is really not helpful in assessments, etc. because it can mean anything from less than 1% to over 99%–basically, anything may happen. I am also surprised that there is any question being raised about impacts being caused (consider the impacts of sea level rise; heat waves; losing the benefits of cold nights to kill infectious disease vectors; landscape shifts that usually occur by over-stressed plants lost rapidly to disease or wildfires and taking many decades to many centuries to become re-established; etc.)—I know of no scientific assessments that raise any question about there being large, adverse impacts from climate change.

Page 1, lines 3-5: Here is a location to make clear on what basis "gains and damages" are going to be evaluated, what will be compared to what. This sentence says it is evaluating SRM, but is not really what is going to be evaluated global warming with and without both abatement and/or SRM? And here is where to explain that CDR as an exit strategy is not being considered in this analysis, which really has implications for issues of the ethics of imposing SRM on future generations, etc. [In this regard, it needs to be said this is just considering quantifiable economic aspects, and this leaves out a lot of aspects, such as issues like loss of biodiversity, inundation of low-lying islands and associated loss of cultures, and so on.]

Page 1, line 6: IPCC, in all of its volumes, I think forbids use of the word "should" as

policy prescriptive and generally scientists and experts considering limited aspects of issues (e.g., here, only considering economic aspects, and quite limited in that regard). It would be straightforward to simply change "should therefore be" to "merits being"

Page 1, lines 6-7: While true as written, there is inadequate context here as long-term CDR, which could well serve as an exit strategy, is not included. One could perhaps say this is included in Abatement, but this ends up depending on what the temperature target is, a point on which there is some debate.

Page 1, line 9 and lines 12-13: While the Paris Agreement set 2 C or preferably 1.5 C as the target, this was not really a scientific choice—it was a political compromise on what was conceived as perhaps possible. Earth's climatic history suggests that the equilibrium sea level sensitivity may be as large as 15 to 20 meters of sea level rise per degree C change in global average temperature. The time lag for equilibrium to occur is probably something like 1000 years plus or minus a factor of 2 (so 500 to 2000 years), and that is perhaps a one-sigma spread on the low side given how ice calving can occur (see paper by DeConto and Pollard). There are those out there saying the real target to meet the objective of the UNFCCC would more appropriately be less than 0.5 C, or 300 ppm, or something similar, and it would seem to me that this paper would be strengthened if it considered the situation for a range of temperature targets (say 0, 1, and 2 C, for example). On the issue of objective of the Paris Accord, there was no decision included in the agreement, as I understand it from one of the lead negotiators, on whether the 1.5 to 2 C was to be a peak and then the temperature would be reduced, or whether these values would be considered as a new level for the future, as is presumed in the IPCC special 1.5 C report, with the thought being this would meet the UNFCCC objective. As the recent IPCC special report makes clear, staying under 1.5 C would require going to net zero CO2 emissions within a few decades, which the report indicates is technologically possible, but quite unlikely politically and economically. For context, it would really be helpful to indicate what the model suggests would be the costs of a long-term steady state global warming of

1.5 or 2 C—and in doing that to make very clear what is included and what is not (presumably biodiversity loss, sea level rise, extreme events, etc.).

Page 1, lines 14-16: As to the set of approaches, there is no mention of aggressive cutting of emissions of short-lived species as a way of delaying warming by a couple of decades and no mention of aggressive CDR (or any CDR), and there are those out there hoping that CDR can be really phased up. The sentence here seems to slightly over-simplify the set of options.

Page 2, line 6: So, "associated changes" is all to be said of the impacts from ongoing global warming? Nothing about what this entails and how severe it could be or about what is and is not included in the calculations that are included, to be included, in the modeling here. I'm sorry, but if there is going to be discussion of, as on line 8, that not all changes in precipitation patterns are reversed, it really needs to be said how much the changes in precipitation are without SRM. There is not even a reference here to the impacts of ongoing global warming. In my view, it is essential to be describing the impacts and giving some indication of levels of confidence, etc. Perhaps show the "burning embers" type of diagram to show how serious the impacts are for context in considering whether the SRM shortcomings are significant or not.

Page 2, lines 7-10: Here is another example of not providing a comparison about how important the inadequacy is. For example, how does the supposed difference compare to the departure that would occur without SRM (so with only the CO2 enhancement)—is the return toward the original perhaps 90%, is the decrease with respect to preindustrial or the CO2 enhanced climate? There are all sorts of SRM injection options yet to be considered such as in amounts and varying by season and latitude yet to be considered, so some tuning of the injection patterns might improve the situation. Are the failures to return to near preindustrial out over the ocean (where there is no impact) or over the land (where there might be)? How does the change compare to the range of natural variability at each location—is it significant (e.g., one can get large percentage changes in precipitation over deserts where the absolute amounts are inconsequential)? I just think the statement here is very inadequate and potentially quite misleading and even pejorative. The real key benefit of SRM might be to reduce sea level rise, but that impact is not even considered. Yes, it is true that ocean acidification is not alleviated, but it needs to be said that the failure to abate rapidly enough will lead to all sorts of impacts, some being irreversible. In my view, this paragraph is just inadequate at presenting what the important situation being addressed is all about—this is where to be indicating the real significance of the issue, and the text here just does not do it.

Page 2, lines 11-12: There also needs to be a comparison to the cost of abatement—where is that?

Page 2, line 13, first phrase: With respect to "moral issues", there are also all sort of moral issues without doing SRM—and these are not mentioned at all. This seems very imbalanced here—SRM would presumably be limiting loss of species due to warming and reducing the rate of sea level rise. I do agree that what is said here is representative of some of the media discussions of the issue, but this is supposed to be a paper that really looks at the tradeoffs, but instead does not even talk about the various impacts of global warming and the ethical and other associated issues.

Page 2, lines 13-14: On the impacts of sulphate, I think a subsequent report by the Robock group has showed that the sulphate coming down from the stratosphere is an order of magnitude smaller than ongoing near-surface emissions and that the sulphate coming from the stratosphere is much more spread out than the near-surface emissions, so at a lower loading and so the impacts are very small. In any case, if the paper is going to talk about health impacts, these need to be compared to what would be the case without SRM, and given the IPCC reports on projected health impacts (not even referenced or described here), the overall health effects would arguably be quite small as compared to 2 to 3 C warming scenarios. It is simply essential in the type of analysis this paper is doing to make comparisons and provide context, so considering global warming with and without SRM and not just talking about SRM impacts in isolation.

Page 2, lines 15-17: Yes, perhaps, but you did say the cost of SRM is virtually trivial, so how about considering whether some other nation or set of nations might take it up if the costs would be so large of the discontinuance. Now, a critical issue associated with this is also if one is proposing that SRM go on forever (though the approach here discounts long-term impacts away) versus there being CDR that could provide an exit strategy for SRM. The really serious governance failure is the one going on in governments not carrying out adequate abatement, which is far more costly and so much more likely, it seems to me, to continue to be done too slowly. For a balanced presentation, context is needed here providing an indication of the relative likelihood of success and failure of each approach (i.e., abatement and SRM). I would also suggest that this paragraph does not provide adequate and balanced context for the comparison that is being carried out in this paper.

Page 2, lines 18-19: First, this is really a comparative analysis of SRM and abatement, not just of SRM. Second, try to avoid the word "should." On the issue of "tipping behavior", some explanation is needed; namely that there are likely multiple tipping points, that some may be irreversible, that some have lag times, and so on—basically, the situation is quite complex.

Page 2, lines 19-20: On the potential for failures of SRM, there is a potential for inadequacy, especially as the temperature to be offset rises to above 2 C, for example—it is a bit hard to see how there could be a possibility of insufficiency for lower temperature increases given that volcanic eruptions do cool the climate. On the issue of the potential for damaging side-effects, again, there is no context here—so this needs to be compared to the type of damaging impacts that would occur without any SRM. This is not an analysis of doing SRM or not—if that were the case, the evaluation would be quite different. This is, however, an analysis of global warming with and without abatement and/or SRM—so global warming is the underlying baseline, and that will have very serious consequences (after all, they are seen as severe enough the nations of the world have been convinced to work to forego any use at all of fossil fuels).

Page 2, lines 20-21: On use of the DICE model, it seems to me that some explanation is needed. What it focuses on, as I understand it, is GDP(or equivalent) without really subdividing whether the expenditures are going toward improving public welfare or are going for costs to adjust and recover from disasters [in one early study of the Social Cost of Carbon, the early DICE model , so seemingly the one used here, apparently calculated that a 10 C global warming, something never experienced on Earth, would only lead to a 30% or so hit on GDP; well, perhaps, but what share of the economy would not be devoted to improving health and welfare]. In my view, the authors need to do some explaining of what "optimal" means in the context of the study being done. There is also the need to indicate the time step and how extreme weather, etc., which typically causes the most damaging impacts, are or are not treated (the random variation, for example).

Page 2, line 31: Is DSICE updated at all from Nordhaus (1992) other than to allow statistical treatment? So, are climate impacts also statistically treated for CO2, or just for SRM?

Page 3, Table 1: Given the page offers extra width, I'd suggest a bit more explanatory phrase under the "Meaning" column. On the various specifics a. What does the value for CO2 forcing mean? Presumably it is not for CO2 doubling as is too high for that. b. What is the basis for the sulphate scaling and how are the three variables different? c. How are b1 and b2 different—there is room to explain? d. How are the two taus different—expand on meaning? e. On pC, why is the value negative? Total precipitation goes up with CO2. f. On economic damage from CO2 concentration, from warming, from precipitation change, from SRM and from SRM, the units all seem strange. Where is the unit for economic value? g. On the implementation cost of SRM, this value seems to be a good bit higher than page 2, line 11. h. Is the rate of capital appreciation, is this the discount rate?

Page 4, lines 2-8: There is no mention of tropospheric ozone, which is responsible for about as much forcing as methane. On line 3, why does this say "industrial aerosol"

given there are other sources as well (so there are really "aerosols")?

Page 5, line 5: Why does this say "industrial processes"? At least in some locations this would mean emissions from industry and not include most transportation, home heating, etc. Perhaps say instead "combustion of fossil fuels".

Page 5, line 6: So, the analysis does not consider any reduction in the rate of C loss from the biosphere? This seems rather strange, given the attitude it will take to do abatement, etc.

Page 5, lines 19-20: Does this mean that one cannot go below a CO2 concentration of something like 400 ppm, so CDR to take the concentration back toward 300 ppm cannot be considered?

Page 6, lines 11-13: This explanation needs to be re-phrased. Over the ocean, the atmosphere does not warm faster than the ocean—the atmosphere has a negative balance and so is warmed by convection up to match the warming of the ocean, which is slow due to its heat capacity. Over land, with a reduced surface heat capacity, warming there can be a bit faster, so the global average temperature of the system can lead the ocean by a bit. Despite this, global precipitation goes up with overall global warming, not down.

Page 6, lines 13-15: I don't understand what is meant that "For SRM, the instantaneous response is weaker." The temperature goes down quite quickly after a volcanic eruption—the reasoning here really does need to be explained more clearly. And how rising CO2 leads to less precipitation is just not clear given precipitation goes up with global warming. Now, it is true that evaporation goes up as well, so perhaps this is referring to available soil moisture, but I am perplexed by explanation here.

Page 6, lines 16-17: What is a bit confusing here is why there is not, perhaps, a non-linear relationship of impacts from changes in precipitation. Basically, there is natural variability and the systems tend to respond, and impacts occur as the changes become

larger, etc.

Page 6, lines 19-21: So, contrary what I said earlier, does this man that damages don't count in the GDP, or is this the effect of there just being less economic activity and responses to other types of damage are included in the GDP? It seems to me here that some clarification is needed—what seems this is talking about is not damage and impacts on people, but only reduction in economic activity. So, if a wildfire wipes out a town and then insurance pays to rebuild it, there is an increase in local, at least, economic activity even though there have been disastrous consequences. Calling this a "damage function" seems rather misleading to me.

Page 6, lines 22-24: On tipping points, this notion that economic impact could only drop to 0.9 seems rather optimistic. Roughly a half dozen countries provide something like 90% of the grain exports in the world that supply of order a 100 nations; we've experienced mild crop failures in 1 of the half dozen nations at a time and seen significant changes—as extremes increase, the likelihood of failures in 2 or more of the exporting nations is likely to go up sharply [Hansen et al. have analyzed observational data of hot summers in Northern Hemisphere land areas, and since the mid-20th century, what were three-sigma warm summers (he divided the land areas into 5-degree squares, so roughly one in a 1000 event) are now occurring over 10% of the time, a factor of over a 100 increase. A severe food crisis would pull substantial funds out of the rest of the economy and the heavily leveraged economy could come crashing down far more than the banking crisis. And then there is what would happen if a large ice stream really collapsed and cause a meter or more sea level rise in less than a decade; the flooding and refugee crisis would be enormous. The real challenge here is that the world does not live based on slow and steady change, but from not much happening to some extreme, and extreme conditions are becoming much more likely.

Page 6, line 26: Why is SRM cost linear? Would there not be savings as more and more is needed (offset perhaps by effectiveness decreasing as get higher and higher—but effectiveness is separately treated)?

Page 6, lines 29-30: I'd suggest that the goal will be to avoid taking the oxygen molecules from the ground to the stratosphere as there is plenty of oxygen up there. So, what is taken up might be pure S or H2S, etc. rather than taking SO2 from the surface to the stratosphere.

Page 7, Figure 1: I don't understand why CO2 is shown to decrease precipitation? Yes, instantly there is a slight stabilization of the troposphere, but that leaves the heat to warm the ocean, and ultimately the increase in the CO2 concentration leads to an increase in global precipitation. What would seem to be needed is to have arrows of different breadth as ultimately the precipitation amount goes up. On the 30% effect of precipitation on economic damage, I guess that might be viable if there is an understanding that the rain is coming down more in extreme events (and it might be useful to mention this).

Page 7, caption to Figure 1: Where does the last sentence of the caption come from? Overall, the injections are strongly limiting warming and so greatly reducing impacts (assuming warming causes significant impact) and one might say one would only count 80% of this, or does this mean that offsetting 2.5K has an overall negative impact. I'm confused.

Page 7, line 4: So, this equation does not seem to account for sea level rise—how is this accounted for or is it not? Are any impacts related to environmental refugees considered and the disruptions that would result? Do the impacts go down over time based on an assumption that adaptation will occur?

Page 7, line 5: What are the consequences of the CO2 concentration change—is this an indication of ocean acidification? Is it the loss of nutrients in food crops from the higher CO2 concentration? Etc. It would help to give an indication.

Page 7, lines 6-7: On this issue of both positive and negative precipitation changes causing negative economic impacts, so what is being compared to what—what is the baseline? So, if SRM leads to much smaller precipitation changes than does global

warming without SRM that its economic costs are less than for CO2, or is the cost based on the departure from the elevated amount of rainfall from global warming?

Page 8, line 6: Is the sea level impact here referring to storm surges? If so, that is not really a complete representation of the consequences of sea level rise given the size of the temperature increase that is being considered—the impact costs of sea level rise will be far, far more than linear.

Page 8, line 15: As previously noted, a really serious but plausible extreme could, it would seem, have much more than a 10% effect. What is the basis for making this the limit instead of having some relationship allowing for smaller likelihoods of much larger economic effects, etc.

Page 8, line 18: How is it that it is presumed that there is zero chance of a tipping point for global warming less than 2 K? This seems very misleading. The Arctic is going to really suffer with global average warming less than 2 K, especially given all that is happening at global warming of 1 K. It would be perfectly impossible to have a food crisis at well below 2 K. And most important, warming of less than 2 K seems quite likely to have started destabilization of several of the Greenland and Antarctic ice streams that would cause long and irreversible sea level rise. I just don't think the 2 K value is justified even though I recall that DICE might have had the assumption of relatively small impacts for less than 2 K?

Page 8, lines 21-22: I'd suggest that the likelihood of unforeseen dangers from SRM keeping the global average temperature near where it is or was in the recent past are far less than the likelihood of surprising outcomes as the global average temperature rises to 2 K and above, a climate regime we are not at all familiar with (and as the relatively greater responsiveness of the Greenland and Antarctic ice sheets are showing compared to what IPCC assessments through the Fourth assessment were indicating).

Page 8, lines 24-25: With SRM failure, does not the temperature go up and that would be expected to lead to very serious impacts, due to both the rate and magnitude. So, is

this sentence just saying there is nothing other than this? If rapid warming rather than gradual warming is the result, would not there be an additional consequence of SRM failing. The abstract hints at this result—so how come it is not part of the calculation here.

Page 8, lines 26-27: I'd suggest making much more of this presumption that the impacts are expected to be the same for everyone in the world, and so there is no consideration of especially vulnerable populations even though virtually all assessments note that the impacts will not be felt evenly. It would seem this needs mention and a reference.

Page 9, lines 1-6: So, sea level rise impacts really don't matter, nor do long term impacts of ocean acidification on the marine food chain? This seems quite perverse, making it such that initiating very significant sea level rise ahead ends up having virtually no present value—no counting of flooding low lying island nations and low-lying coastal areas, no treatment of loss of biodiversity, and so on. As noted in the general comments, I'm very troubled by how the implications of various of the assumptions are just not discussed—this just has to be done to show the limits of the study so future studies will be done better. It seems to me that the way to do this calculation would be to have some share of the impacts with a discount rate and some not as there is simply no way to replace the Amazon rainforest, etc.

Page 9, lines 20-21: It would be best to avoid use of the words "may" and "might" and say something like "a thorough analysis and evaluation would consider not only the ensemble mean, but also the full range of possible values."

Page 9, line 29: Is there not a fourth scenario of doing nothing at all? That would seem to be needed to get a sense of how serious unaddressed climate change would be.

Page 10, line 6: How is that it will take until 2055 to even start SRM? That seems far too long. Indeed, it would seem that waiting until the global average temperature has risen to over 2 K before doing anything. Ideally, it would seem, one would start

[Figure]

ASAP and do SRM at a slowly increasing amount to avoid the irreversible impacts of going up to over 2 K before acting. Unfortunately, however, the analysis here seems to not really treat irreversible impacts and situations where long-term impacts with a very large hysteresis like major loss of ice from the Greenland and Antarctic ice sheets are initiated. Also, assuming only 30% probability—is that due to technological issues or governance issues, etc.? I just do not think the assumptions here are particularly plausible.

Page 10, lines 14-15: So, is the demand for electricity for air conditioning reduced with SRM? A study I know of says this is a really large term—namely that SRM slows the rate of global emissions of CO2.

Page 10, line 16: Is there really enough carbon around to go to 2000 ppm of CO2, especially at the rate shown so that it would be continuing to increase beyond that? This seems a very extreme base case to be running. I would very much like to understand the impacts that would be generated by the real baseline case of CO2 growth with no abatement and no SRM.

Page 10, line 19: What is the comparison being made here. Is this saying that the damage of injection of sulphate is higher than the impacts, which would presumably near zero as there is no temperature increase? The real comparison that needs to be made is the impact cost of sulphate to the impacts that would be resulting at that time from global warming of 5.4 K in 2400; with that level of warming, virtually all of the ice sheets would be melted and sea level would be up by 50 meters or so, and the damage due to that would, I would think, be far, far larger than the impacts of the sulphate injection (well, GDP might still be viable, but it would all be going to relocating the global population and not to enhancing human welfare). And then global agriculture would be very seriously disrupted and all global ecosystems would have been destroyed. I just do not think this sentence has been at all justified.

Page 11, Figure 2: I don't understand the SRM (lower right) figure. If the assumption is

that SRM cannot begin until 2055, how come this graph shows the green line jogging down right after 2015. Is this saying we are not now doing what makes sense in terms of abatement—if so, this could usefully be mentioned. With respect to the upper right diagram, why is this the only figure that starts in 2005 instead of 2015?

Page 12, Table 3: The value of the SCC looks very low—and it is not clear when this value applies (is this a starting or ending value?). If the SCC is really this low, it really seems that the estimated impacts are far less than is being discussed in international assessments (and this was a characteristic, as I recall, of the original DICE model). I'd also like to better understand how the abatement costs are calculated—with solar and wind costs already ending up lower than coal for electricity and solid state batteries coming along that will greatly reduce the cost of converting the transportation sector, is the abatement cost taking into account the rate of technological change going on and that would be encouraged by a rising SCC? This actually raises the question if there has been any verification of the DICE model for some period, say, 1970 to 2020 or something. It would be nice to understand what the confidence level for the DICE model is.

Page 12, lines 14-15: This is a very strange way to treating tipping points. There are likely multiple ones and the idea that going to greater and greater warming would be more acceptable after passing one tipping point seems to be a serious shortcoming of this model.

Page 13, Figure 3: So, figures g, h, i show a tremendous slippery slope problem (which is eventually discussed on page 14, lines 25-30). This is, I would presume a result of fossil fuels having a lower cost than the alternatives. Given that the crossover to less expensive renewables is occurring now for electricity generation and I would argue will happen within two decades for electric vehicles (once solid-state batteries replace lithium batteries), so why in the world would people go back to fossil fuels as SRM comes along somewhat later. I just do not understand what is leading to this unless technological improvement is not occurring.

Page 14, line 7: This is really bizarre that SRM is not allowed to start until T equal 2 K. I presume this is because of the limit imposed that SRM cannot start before 2055. One would think that starting earlier and with less intensity would make the most sense and would avoid irreversible losses to ice on land, biodiversity, extreme weather impacts, etc. out until that date. So, the discussion here is all because of an assumption made on when SRM can be started. Such presumptions need to be stated and then stated again when it determines a result.

Page 14, line 16: This notion of SRM failure is quite strange.

Page 15, Figure 4: While the IPCC 1.5 report does presume that the Paris temperature goals meant these could be considered a new, non-dangerous, steady state, paleoclimatic data simply do not support this proposition unless one is willing to tolerate of order 50 meters of sea level rise. Also delaying SRM until 2055 leads to some impacts that one would like to avoid by starting SRM earlier. Allowing the CO2 concentration to go so high seems likely to lead to an unacceptable level of ocean acidification, especially for coral atolls, and quite possibly global shellfish productivity and more. It seems to me the economic damage function used does not come close to representing the actual impacts that will be occurring. It seems pretty clear to me that using the DICE model avoids getting at the actual impacts on the environment and society, and so global human and environmental welfare as opposed to just the drop in GNP. Again, it is absolutely essential that the shortcomings of this analysis be indicated, even though the ultimate conclusion of doing everything one can makes the most sense.

Page 16 line 1: Imposing 2055 and 30% probability is quite limiting. It would be interesting to see a sensitivity study on these values, but since the damage function is only related to total economic activity and does not treat irreversible losses or initiating long term ice sheet loss, etc., probably a small effect. Very clearly, the limitations imposed by the use of the DICE damage function has a very limiting effect on the analysis here.

Page 16, lines 2-3: In saying "identify unacceptable environmental risks", there needs

to be context—so compared to what would happen without SRM and with only abatement, etc. This establishment of some absolute side effect that is unacceptable seems inappropriate to me—one has to provide context for the decision. Policymakers make decisions on relative likelihood or balance, and so will want the comparison.

Page 16, lines 5-6: So, this sort of assumes, one failure and never use SRM again, even though the impacts of very large warming would be very significant. So, it needs to be stated this is presumably an irrevocable decision despite other impacts.

Page 16, line 17: So, most of the abatement going on will be a result of a technological advance. Once a technological advance becomes economical, it will likely go forward independent of whether SRM is coming or not. I guess the decision-maker could reduce a subsidy and that might slow a transition, though this really means one is not allowing at all for movements, even though movements would seem to be a major factor in transitions rather than just economics. Another limitation that needs to be restated, etc.

Page 16, line 26: I think it needs to be explained why the SCC is so much lower here than has been estimated in other studies. With the international leaders seeing 2 K as essentially unacceptable, the Shadow Price of Carbon (so the value to keep below 2K) is pretty clearly much greater than the SCC value—indeed, the SCC value here seems much lower than is generally being discussed. I think the authors need to indicate the reasons that the SCC here is so low compared to what has been shown in other studies of the SCC.

Page 16, line 31-33: In that SRM is based on physical mechanisms (so no chance of a runaway effect as could happen were there a biological factor being introduced in the environment) and for which there is a pretty close natural analog (namely volcanic eruptions), the likelihood of "huge unexpected damage" would seem to be pretty small—and in mentioning this here, there again needs to be context provided, so huge compared to what? On the need to continue SRM research, this conclusion would

seem to merit much greater prominence.

Page 17, line 1: Generally, I like the Sensitivity Analysis section. I would be interested in seeing a sensitivity study on the rate of technological advancement.

Page 17, line 9: I am confused; SRM will surely have greater benefits than negative impacts, otherwise why would it be continued? Yes, there may be larger or lesser benefits, but how is it that SRM will cause net damages [I guess, if one counts only dollars using the specified discount rate, one may get a negative consequence if SRM is kept going for reasons other than its effect on the overall economy (e.g., to preserve biodiversity that is not in the damage function)].

Page 18, line 3: What about also halving the damage associated with SRM.

Page 18, lines 21 to Page 19, line 3: This is just far too far back in the document and far too limited to be an adequate disclosure to the reader of the shortcomings and limitations of this study. Yes, it is a start, and the conclusion may be robust, but there is much, much more to be done and far too many limitations and failures to enumerate the shortcomings. Such disclosures need to be made clear up front, mentioned throughout, and included and evaluated in the summary and discussion section.

Page 19, lines 5-7: Putting all the emphasis on uncertainties relating to SRM is just not a balanced conclusion; consider issues of the base calculation not treating sea level rise, irreversible losses like biodiversity, and so on—the DICE damage function is just not adequate for the study undertaken except in the most general sense.

Page 20, line 30 and page 21, line 1: Just a comment that it is quite easy to warm the planet—just add some CFCs, or HFCs if you want to have a shorter lifetime. Geoengineering warming is far easier than intervening to cool the system. There are issues of time constants, etc. but avoiding an ice age would be far easier than creating one.

Technical Comments

Page 1, line 1: Suggest changing "in spite of" to "caused by"

Page 1, line 8: Clearer if change "can" to "can only"

Page 1, line 14: While Crutzen (2006) did get the recent discussion going, the way the text reads it is a bit hard to understand how a 2006 reference is a response to the Paris Accord, as the text on lines 12-14 seems to convey.

Page 1, line 20: I'd suggest changing to "stratosphere, which would, like an ongoing volcanic eruption, increase"

Page 1, line 21: The reference to "Thomson et al." has a different spelling in the references.

Page 2, line 4: "sufficiently strong forcing can be achieved" to do what—go over 2 W/m2; if so by how much.

Page 2, lines 23-24: In references, "MorenoCruz" is hyphenated. On line 24, "which" to "that"

Page 8, line 2: Not clear what this is referring to, and why is the effect here 20% and not the 10% in Figure 1?

Page 10, line 18: "very high injection rates"—that you are referring to injections of sulphate (actually, of course, earlier the assumption was that it was SO2 being injected, not sulphate).

Page 12, line 10: Change to "individual"

Page 13, Figure 3. Why does the upper left diagram go to 2400 instead of 2300 like all the other figures? On figure j, k, i, I'd suggest having the same vertical span for all three figures so one can more easily compare results. For figures g, h, I, the word "Atmospheric" is misspelled.

Page 14, line 8: I don't understand what a performance of "181%" means? What is

being divided by what?

Page 14, line 9: The graph only goes to 2300, not to 2415? Going out that far seems very questionable. What one really wants to do is start SRM early and then have CDR come on so it can be phased out.

Page 20, lines 21-22: I don't understand the meaning of the phrase after "with"

Page 22 ff. on references: a. In Cai, spelling is "Stanford" b. In Cai et al., why is article title in title case? c. In Keller et al., what is "ncomms"? d. Stowe et al, is out of alphabetical order and initials of first author need to go after the name

Please also note the supplement to this comment:
https://www.earth-syst-dynam-discuss.net/esd-2019-5/esd-2019-5-RC1-supplement.pdf

―――――――――――――――――

---

## Referee Comment (RC2) · Anonymous Referee #2 · 23 Feb 2019

This article makes a very nice start towards seriously integrating solar radiation management (SRM) into an economic model of climate change. A policymaker can choose to reduce CO2 emissions or to inject aerosols into the stratosphere. The former option reduces future CO2 and temperature, whereas the latter only reduces temperature. The key extensions to the benchmark economic model of climate change are a model of the relation between SRM, temperature, and precipitation, a decomposition of climate change damages that distinguishes SRM from CO2 emission reductions, and a dynamic programming formulation (borrowed from other literature) that allows for optimal policy to account for uncertainty about SRM failure and about a type of permanent economic loss that the authors refer to as a tipping point.

[Figure]

It is kind of obvious that the policymaker will choose to use both SRM and emission reductions. The potentially interesting conclusion is the relative mix, which the authors report includes modest SRM. However, this conclusion is likely to be sensitive to hard-to-pin-down assumptions about the cost and damages from SRM (as well as the damages from CO2 and the costs of eliminating it). It is unclear how big of an advance this result is relative to previous literature (Heutel et al). I would prefer to see a more insightful conclusion and will offer suggestions to that effect.

First, a more interesting question about SRM is, to me, not the precise level at which we should be using it now but whether we should start using it at all. The current model has no force that would lead a policymaker to delay introducing SRM. I can think of two relevant forces. First, there may be a fixed cost to beginning SRM, whether political, economic, or ecological. Second, the authors assume (I think) that the risk of SRM failure is constant over time. A more realistic model would have a small chance of new research finding a fatal flaw in SRM as long as it is not used and a much larger chance of actual consequences revealing a fatal flaw once it is used. I am suggesting that the probability of failure should be low as long as SRM is not used and then be higher once it is used. A policymaker might then choose to delay beginning SRM until it is really needed, since SRM may not last for long once it is begun. I strongly encourage the authors to explore these (or other) ways of making the paper about when to begin using SRM.

Second, the interesting aspect of SRM value is, to me, its insurance value. The authors analyze a dynamic programming problem, so they have the machinery to answer that kind of question, but they fail to get there. Instead, they analyze perhaps the least interesting form of "tipping" that they could have. Their tipping point permanently reduces economic output, but the interesting aspect of SRM in a world with tipping points is the potential for SRM to manage a tipping process that is underway. Had the authors chosen to model tipping points in the way that Lemoine and Traeger (2014) did (where a tipping point is a sudden change in a parameter of the physical system,

which manifests itself in temperature and economic output only over time and in a fashion that depends on subsequent policy), SRM could play a key role by allowing the policymaker to intervene after triggering a tipping point so as to mitigate the consequences of that tipping point. Alternately, had the authors studied uncertainty about future warming, SRM could have been used to control the consequences of ending up in a high-warming world. Finally, if the model allowed for the more interesting type of tipping point (or for warming uncertainty) along with a reason to delay beginning SRM, then we may see the policymaker delay SRM until a bad state of the world warrants using it. Learning whether/when that plan is optimal would be interesting.

Smaller comments:

-The authors should spend more space clearly laying out the methodological, calibrational, and conceptual contrasts with Heutel et al. They also should elaborate on the differences in results and speculate as to their origin,

-Technical notes: 1) The authors describe how they mitigate the problem being non-smooth, but it seems far more straightforward and more realistic to simply assume that the chance of tipping is smooth. Assuming no risk of tipping below some temperature is arbitrary. 2) The authors impose an upper bound on injection rates. This is a curious choice. I would prefer the authors to model the source of the constraint. As it is, either the constraint binds or it does not. If it binds, then it is important, the authors should highlight it in their results discussion, and the authors should consider eliminating it. If it does not bind, then it is unimportant and the authors should consider eliminating it (or at worst justify it in terms of numerical convenience but say it is irrelevant in practice).

-Sensitivity checks: I want to see more sensitivity analysis with respect to the fairly arbitrary damage parameters, and especially to the $\psi_S$ that I'm not sure has been explored. This should be the core of the results. I am far more interested in learning about what we need to believe to get near-term SRM to be high or low than in learning about the spread of future policy.

-I wasn't happy with the discussion of precipitation damages in 2.1.3. There are plenty of ways precipitation can matter, independently of temperature. Think of crops. One could have just as easily said that many temperature channels are mediated by precipitation. Plus geoengineering changes precipitation in ways that are not mrely determined by temperature. For instance, might geoengineering change patterns of rainfall (such as monsoons)? Perhaps these kinds of effects could be captured by $\psi_S$, about which very little is said other than that its level is arbitrary.

-The objective in (10) must be wrong. I think the policymaker must be maximizing expected welfare, not just welfare. And in light of that comment, the motivation for the final paragraph in 2.2 doesn't make a lot of sense. A risk-averse policymaker (as modeled here) has already chosen policy in light of the range of possible cases. It is not internally consistent to postulate caring about percentiles of the subsequent policy performance because of risk aversion.

-In s 3.2, the authors report that abatement decreases after triggering a tipping point. This is purely an artifact of the type of tipping point modeled here. It would probably not arise if abatement had any role in controlling the consequences of a tipping point, as in Lemoine and Traeger 2014.

-Why is the scc unaffected by the possibility of tipping in the abatement+SRM world?

-It seems to me that the final sentence on page 16 would be stronger with a comparison between maximized welfare in the "realistic" world and welfare in the same world if the policymaker used the policy from the abatement-only world. How much could today's policymaker gain by anticipating the future possibility of SRM?

-The discounting result in 3.4 was one of the more interesting results of the paper: a higher discount rate favors SRM because it causes damages, may fail, and doesn't curb future CO2. I would suggest highlighting it more and perhaps doing more with it.

---

## Author Response (AR1)

**Answer to Reviewer 1**

June 6, 2019

A general remark:

The reviewer repeatedly suggested describing in greater detail issues which are already treated elsewhere in the literature (most notably concerning the design and shortcomings of the DICE model and impacts of climate change). While we do appreciate his effort to eliminate potential ambiguities or omissions, we also believe that a research paper should not contain too much repetition of existing literature, and therefore tried to address many of these requests by adding a suitable reference.

**General Comment 2:**

To indicate that we indeed deal only with SRM, we did:

- replace "Geoengineering" by "Solar Radiation Management" in the title DONE
- mention in the introduction that negative emissions are not included DONE
- mention in the discussion part that trade-off between negative emissions, adaptation and SRM might be interesting wo study (with a more detailed model) DONE

**General Comment 3 + 4:**

We included in the introduction a short paragraph with references concerning model limitations (in particular: damage function; abatement costs not including learning-by doing; single decision maker; no CDR), but stated that we still believe that DICE to be a useful testbed for exploratory studies, which should of course be expanded by follow-up studies with more detailed models.

We also expanded the part on model imitations in the discussion. In particular, we clarified better how the limitations of the energy sector in DICE impact our results and mentioned the omission of negative emission and adaptation. DONE

However, we did not repeat details of the DICE model, e.g. the construction of the damage function, since these can be found in the cited literature. We did add Nordhaus and Boyer 2000 to the references, because this paper gives a lot of detail on how the damage function is constructed.

**General Comment 5:**

It is hard to compare the uncertainties of SRM and (unmitigated) climate change, because especially the former ones are hardly quantified yet.

The most important pair of options which arises from our study is not "Geoenginereering vs unconstrained climate change" (at least we strongly hope this will not be the decision humanity will face at the end!) but "Abatement-only vs Abatement+SRM (and if so, how much SRM)". So basically, we want to investigate whether we should add SRM to our action portfolio or relate on abatement only. And for this question, the uncertainties surrounding SRM are absolutely vital - and far less well studied than the likewise daunting uncertainties surrounding climate change.

By the way, volcanic eruptions provide only an incomplete analog to SRM because their effect is relatively short-term; and it is highly unlikely that SRM-only will keep the climate "more or less at its present state", in particular as far as precipitation is concerned.

We added a reference to IPCC WG2 for an overview on climate damage (introduction) and briefly mention the issues of irreversibility and delayed damage (e.g. in case of ice melt). DONE

**Specific Comment Title:**

See General Comment 2

**Specific Comment P.1 L.2:**

- "lower" -> "to lower": adjusted DONE

- "may": note that this is about damages from SRM, not from climate change. And there is *huge* uncertainty about adverse economic effects of SRM, so "may" is quite appropriate... no change made.

**Specific Comment P.1 L.3-5:**

This is the abstract, and therefore should be concise. So it is not the place to list everything *not* taken into account in this study.

To clarify that the focus is on economics, we replaced "gains and damages" by "(economic) gains and damages". DONE

**Specific Comment P.1 L.6:**

change "should therefore be taken" into "therefore merits being taken": adjusted. DONE

**Specific Comment P.1 L.6-7:**

Yes, we agree that CRD should be mentioned, but not in the abstract. See general comment 3-4.

**Specific Comment P.1 L.9 & 12-13:**

The paper is not primarily about reaching temperature targets. The statement in line 9 is a descriptive one; it describes a result but we did not enforce any target into the simulation itself. So the suggestion to try several targets is not applicable.

The statement in line 12-13 merely illustrates that humanity is not on its way to reach any suggested temperature target through abatement, which might make additional measures like SRM attractive. It is outside the scope of this study to discuss the justification of 2K or 1.5K as target.

For what is included in the damage function: Please consult the cited literature, in particular Nordhaus.

**Specific Comment P.1 L.14-16:**

Omission of CDR is clarified in introduction (see general comment 3-4)

**Specific Comment P.2 L.6:**

Again, this study is not an assessment of the severity of climate change, so it is not the scope of this paper to list or discuss all climate impacts in the text (as mentioned, we added a reference to IPCC WG2 for an overview).

Here we change "associated damages" into "heating-induced damages", to reduce ambiguity (since some damages associated with / sharing common cause with climate change, like precipitation changes and ocean acidification, are not directly caused by warming). DONE

**Specific Comment P.2 L.7-10:**

Please note that part of the questions raised are treated in the literature cited.

Concerning the expected reduction in global precipitation: I am aware of no SRM study in which tweaking injection parameters (e.g. latitude, height, season of injection) avoids precipitation decrease, and there are plausible physical mechanisms why such a decrease should take place. Concerning precipitation patterns, there is more uncertainty and I expect that this more sensitive to injection parameters than the sign of the global precipitation trend. This is why we used the weak formulation "may": It is quite perceivable that these pattern change (thus an important risk), but model uncertainty and the fact that injection parameters are undecided make it hard to quantity these effects.

Concerning sea level rise, this is one (important) component of temperature-induced damages (see Nordhaus' paper for details).

**Specific Comment P.2 L.11:**

The cost of abatement is highly uncertain, because effects such as innovation, learning-by-doing, and customer behaviour are hard to predict (to say the least).

As a very crude indicator, we consider the cost of installing enough solar panels to meet its current energy demands.

Solar cells cost around 1 \$/W of capacity; this rises to 4\$/W when taking into account installation costs (at household level; may be different for large, remote solar parks). The sun does not always shine; let us be pessimistic and assume that only 20% of the capacity is actually used. This leads to an effective price of 20\$/W of actual capacity. [Numbers from Cassedy and Grossmann; see below for precise reference.]

The current human energy consumption is roughly $1.1 \times 10^{17} Wh/yr$. Installing solar cells to meet this demand would cost $\frac{20\$}{W} \times \frac{1.1e^{17}Wh/yr}{(365 \times 24)h/yr} = 2.5 \times 10^{14}\$$ or about 6 times the world GDP. This is about 600 times as much as the yearly cost of SRM at an injection rate of 30Mt(S)/yr (typical SRM level in our Abatement+SRM scenario at standard model settings).

Obviously, this is a very crude indicator for abatement cost. Our measure does not include maintenance costs of the solar panels (which are currently about 20% of installation costs), nor the much larger costs of dealing with the intermittency of the energy supply (restructuring the energy grid, storage capacity...), or growing energy demand. On the other hand, it does not include the benefits of energy efficiency increase, cost reduction by technology improvement (currently about 20% cost reduction per doubling of installed capacity), reduced cost for carbon-based energy production, pre-existing renewable capacity, etc.

We include this estimate briefly for comparison. DONE

**Specific Comment P.2 L.13:**

This paper is about whether or not to use SRM, not primarily about ethics of CO2 abatement. SRM has moral issues, such as the question of whom to put in charge of it, which are novel compared to those associated with global warming and abatement.

**Specific Comment P.2 L.13-14:**

Concerning the study by Robock on SO2 settling: The study used rather modest injection rates (about an order of magnitude below what we are considering here) and mainly dealt with the effect of SO2 on plants. The conclusion was that likely, at these small injection rates, the damage to vegetation is small, but at least in some regions, damage could occur if injections were tenfold.

Despite having severe impacts especially in large industrial agglomerations, industrial SO2 injections are a relatively regional phenomenon, because they are injected at low levels and washed after about a weak. SO2 from SRM could affect the entire atmosphere. So Robock's study certainly does not prove that SO2 loads from SRM have little effect. While the currently available studies, some of which we cited, give reason for concern, they are not extensive enough to quantify these concerns. Please note that in the Summary and Discussion section, we advocate to investigate these effects with a high priority.

The health issues associated with global warming are assumed to be part of the damage function for warming, and therefore - implicitly - reduced if the warming is reduced. A full assessment of health effects of unmitigated vs geoengineered climate is beyond the scope of this paper.

See also other comments concerning aspects of the damage function.

**Specific Comment P.2 L.15-17:**

While I agree that it is unlikely that the cost burden - or technical feasibility issues - will cause a sudden discontinuation of geoengineering, termination shock is still a relevant concern. (Armed) conflict could be one cause. More importantly, it cannot be excluded that the damage associated with SRM is higher than believed (for example, it could be that after a few decades of deployment it turns out that the long-term-effect on the ozone layer is detrimental, to name just one possibility).

And it is important to make clear that sudden disruption for whatever reasons could be detrimental, and therefore if choosing for substantial SRM, we are committed to continue it or at most phase it out slowly.

As you can see in the result section (fig. 2b, 3c), injection rates indeed decline towards the end of the simulation in the Abate+SRM cases. This is qualitatively the effect you mention, namely using SRM as a transition technology

until abatement (which in principle could include CDR) sufficiently reduces CO2 concentration. We now highlight this important point in section 3.1. DONE

**Specific Comment P.2 L.18-19:**

The main aim of the study is to investigate whether or not to add SRM to the policy portfolio in addition to the already well-studied abatement. Calling it a "comparison between SRM and abatement" would imply that these options are alternatives. But the main alternative is "Abatement+SRM" vs "Abatement only" (or for short: "SRM or not").

We reformulated the "should". DONE

Concerning the explanation of the tipping point: It is numerically very hard to include several tipping points at the same time. We will now also study one different tipping point (namely a positive albedo feedback causing intensification of warming[1]) for diversity. DONE

**Specific Comment P.2 L.19-20:**

"Inefficiency": There are studies (Kleinschmitt 2018) which suggest that the maximum achievable long-time cooling might be limited - which would imply that at least extreme scenarios like cooling "RCP8.5"-like temperatures to pre-industrial values are impossible. Inserted reference. DONE

"Damage": while obviously the damage of unmitigated climate change would likewise be huge, it is not totally inconceivable that SRM causes some (unforeseen) damage, like massive destruction of the ozone layer or something we simply failed to think of. This may be unlikely, but we cannot know for sure. The inventors of the steam engine didn't think of global warming, either... We added the word "*unforeseen* damage" to clarify this, plus a reference to Robock2009. DONE

**Specific Comment P.2 L.20-21:**

Reformulated "the optimal policy" into "the (economically) optimal policy" to clarify this. DONE

**Specific Comment P.2 L.31:**

DSICE was developed based on the 2007 version of DICE, and later on the DSICE framework was adjusted in line with DICE2013. Now clarified and reference added. DONE

The main objective of DSICE w.r.t DICE is indeed to add noise or tipping. This was a major piece of achievement.

In our study, as described later in the methods section, the stochastic elements are only the tipping point and SRM failure.

**Specific Comment P.3 Table 1:**

If one wants to give a detailed explanation of all variables, then the extra width will not be sufficient. Also, the symbols are explained in more detail in the methods sections. However, to ease orientation, we added references to the corresponding equations or sections (or, in one case, to the literature). DONE

Concerning a) The value is the radiative forcing for e-folding (not doubling) CO2; see eq. 1

Concerning e) Precipitation goes up with CO2 because temperature goes up. If you add CO2 but keep surface temperature constant, then precipitation actually goes down. $p_C$ only describes the pure CO2 effect; the temperature effect is in $p_T$. See eq. (6).

Concerning f) Damage is unitless, as it is expressed as fraction of GDP (see eq. 7); the damage associated with quantity $X$ is of the form $\psi_X X^2$. This explains the units.

Concerning g) P2, L11 gives the range of cost per ton of injected gas; we assumed an intermediate value of $7 * 10^9 \$/Mt$. The table gives the cost per ton sulphur. Since we assume the injected gas to be SO2, which has twice the molecular weight of elementary sulphur, costs per ton (S) are twice as high as per ton of injected gas. This was also explained in line 27-30 on p. 6.
* * *
[1]described as tipping point due to methane release in the original response letter, but since it is reversible, this (stylised) tipping point is better interpreted in terms of a purely temperature-dependent albedo feedback

Concerning h) If you mean capital depreciation, this has nothing to do with the discount rate, but with the fact that capital (e.g. machines) looses value over time (e.g. by being worn-out our becoming outdated). This is explained in Nordhaus 1992 to which we now added a reference in the table.

**Specific Comment P.4 L.2-8:**

Indeed, tropospheric ozone is a relevant greenhouse gas and should be mentioned here. DONE

On the other hand, since ozone is a byproduct of chemical reaction with anthropogenic emissions (e.g. CH4, CO in presence of NOx), the argument that "other forcing" can partly be abated, also holds for O3. So the coarse estimate on P4, L 11 can still be used.

Reference: Myhre et al., 2013 (IPCC WG1, chapter 8; see p. 760ff; fig. 8.4 also shows that scenarios with high CO4 concentrations tend to have high tropospheric ozone, confirming that ambitious abatement leads to lower tropospheric ozone)

**Specific Comment P.5 L.5:**

replaced "industrial processes" by "fossil fuel combustion" DONE

**Specific Comment P.5 L.6:**

True, the model does not explicitly include the effect that a reduction in forest area reduces the CO2 sink stemming from Carbon fertilisation (while the direct loss of carbon, namely the loss of biomass and humus, is assumed to be included).

While this effect is surely relevant for detailed analysis of the carbon cycle, I am fairly sure that the effect is not bigger than, for example, the error bar on the total carbon fertilisation effect itself. Simulation the carbon cycle to such a degree of detail is beyond the scope of this exploratory study.

**Specific Comment P.5 L.19-20:**

No, this simply sets the initial conditions. While we are not doing so now, it is in principle possible to include CDR by having negative emission terms in equations 3b and 3c ($E < 0$).

**Specific Comment P.6 L.11-13:**

Yes, global warming increases global precipitation. However, the atmosphere not only warms due to heating from below, but also from absorbing long-wave radiation. This is explained in detail in the studies by Andrews 2010 and MacMartin and Kravitz 2016 (especially their fig. 2b) cited here.

If you suddenly increase CO2 (say, double it instantaneously), then you get a negative precipitation response at first (timescale of about a year) and only then, when the surface warming effect kicks in, precipitation increases. Likewise, when you both CO2 but keep surface temperature constant (say by SRM) then precipitation also decreases.

If you increase CO2 gradually and have no SRM (i.e. the current realistic conditions) then the warming effect is stronger than the direct CO2 effect, leading to net precip. increase when increasing CO2. We now pointed this out in the text. DONE

**Specific Comment P.6 L.13-15:**

This sentence (line 13) is about (direct) precipitation response, not temperature response. The explanation here was merged with the previous one (just prior to eq. 6) and slightly expanded, to generate more clarity. We also explained the meaning of $p_T T$, $p_C F_C$, $p_S F_S$ here. DONE.

**Specific Comment P.6 L.16-17:**

While non-linear effects might be present, MacMartin and Kravitz show that a linear emulator yields quite a reasonable approximation. So there is no use here to introduce additional complexity.

**Specific Comment P.6 L.19-21:**

Since we are taking over the damage function approach from DICE, we will not repeat its explanations here, but added the corresponding reference to Nordhaus' work.

Nordhaus tried to incorporate non-GDP aspects into his damage function (see also Nordhaus and Boyer 2000, now cited in the introduction and section 2.1.3), for example through his "willingness to pay" approach. The DICE damage function has been criticised extensively, and with good reason. However, creating a proper (or even acceptable) damage function is tremendously complex and still an ongoing field of research. So this is outside the scope of this exploratory study.

In your example, the loss of the town would count as a damage in Nordhaus' approach. Basically, the idea is that each period, humans produce "something" ($\bar{Y}$, the gross GDP) and part of it gets lost due to climate change. Non-material losses are converted into material loss.

**Specific Comment P.6 L.22-24:**

10% economic drop *forever* is quite significant (for comparison, the Great Depression around 1930 cut GDP by 15% for just a few years), and of the same magnitude of losses used in other studies, like Cai and Lenton 2016. Of course, you could come up with scenarios that are worse. However, doubling the tipping damage does hardly affect policy (see sensitivity run in table 4). A reason is that tipping usually occurs in the far future, which is strongly discounted. A second reason is that tipping is unlikely, if abatement and SRM are combined (i.e. it only occurs SRM failure).

In general one might question the DICE model's optimism about economic growth, and the way how discounting is used. This is outside the scope of this study.

**Specific Comment P.6 L.26:**

There would likely be initial development costs which don't increase with the amount of deployment.

However, the manufacturing of airplanes, fuel costs, costs for producing the injected gas, and manpower for operation all increase probably linear with injection rate. Given the large uncertainty (factor of about 5) in the cost estimates, we decided to stick to a simple, i.e. linear approach, and omitted initial costs, for which there is no good estimate.

Decreasing effectiveness of SRM is indeed treated separately, namely through the radiative forcing equation.

**Specific Comment P.6 L.29-30:**

It has been considered to use H2S. However, some raise concern that this gas is much more toxic than SO2. Also, we wanted to be conservative, and prefer to overestimate rather than underestimate the cost. This is now clarified in the text. DONE

**Specific Comment P.7 fig1:**

The pathway CO2-warming-more rain is represented by the two arrows with a plus sign going from CO2 to T and from T to P. The arrow from CO2 to P is only the direct effect. The direct effect is quite important in the context of SRM which can partly offset the pathway via warming. Although the temperature-mediated effect of CO2 on precipitation is dominant in case of slow CO2 increase and zero SRM, it needn't be dominant when SRM is involved. So trying to scale the arrows could be misleading (especially if you want to do it in a quantitative way, e.g. letting the arrow width scale with the strength of the effect represented, because the corresponding constants have different units).

However, we inserted a reference to eq. 6 in the figure caption, so that the interested reader can refer back to the (now improved) explanations. DONE

**Specific Comment P.7 fig1 caption:**

The mentioned damage from SRM is only the damage caused by SRM directly; reduction of damage from global warming is of course also modelled, but this goes via reducing the warming $T$. We inserted that SRM causes "*direct* damage of 20%" to clarify this. DONE

**Specific Comment P.7 L.4:**

Sea level rise is accounted for as part of the temperature effect.

Adaptation is not included (yes, one of the many shortcoming of the DICE model... now mentioned in the introduction).

**Specific Comment P.7 L.5:**

Atmospheric CO2 may not be damaging in itself, but we use $C$ as a (rough) proxy for ocean acidification, which we do not model explicitly. This is now clarified. DONE

**Specific Comment P.7 L.6-7:**

The baseline for $P$ is pre-industrial (as mentioned two lines above, L5).

The baseline for damage is also pre-industrial (in other words, if $P, T, C$ would remain at pre-industrial levels, no "extra" damage from climate change occurs) which followed from the fact that $D = 0$ if $C = 0, T = 0, P = 0$.

I do not fully understand the last sentence. If the precipitation changes with SRM are smaller than without SRM, then the contribution of precipitation to the damage is smaller with SRM than without.

**Specific Comment P.8 L.6:**

The cost of sea level rise (which also makes storm surges more severe) is included in the temperature contribution of the damage (see Nordhaus and Boyer, 2000). That contribution goes quadratic with temperature, i.e. not linear (See eq. 8)

**Specific Comment P.8 L.15:**

See also previous comment on tipping.

The constraint is partly numerical. You could in principle allow multiple tipping points, for example one with higher likelihood and 10% loss, and one with lower likelihood and much higher loss. However, with dynamic programming, this will increase computational efforts very much, and given the relatively small impact of tipping on overall policy, we did not do this. (Cai et al, 2016 did, but they had only one decision variable, namely abatement.)

To add diversity to the representation of tipping points, we are now also investigating a different type of tipping point, in which global warming triggers a positive warming feedback (suggestion by reviewer 2) thought of as albedo feedback.

DONE, new sect. 2.4

**Specific Comment P.8 L.18:**

Some tipping points are considered possible at lower, some at higher thresholds than 2K. Since we have included only one stylised tipping point (see previous point), we chose a compromise. The value is inspired by the Paris agreement to keep (well) below 2K warming. We now performed a sensitivity run with 1K as threshold, but the effects on policy are minimal.

DONE

**Specific Comment P.8 L.21-22:**

The idea here was to investigate in a stylised, qualitative way the hypothetical possibility that SRM might in whatever way be "unreliable". Being conservative towards geoengineering, we preferred to overestimate this possibility rather than underplay it (since no good estimates are available anyway).

Disruption of geoengineering is not totally inconceivable (who knows, it *might* destroy the ozone layer, or it *might* be that a war destroys the SRM infrastructure).

Since the result is that SRM is used anyway, despite the rather large failure probability (see also section 3.4, sensitivity runs), choosing a smaller failure probability is unlikely to affect the overall results.

**Specific Comment P.8 L.24-25:**

The matter of the termination shock is briefly investigated in section 3.3. We now added a remark here to clarify that the matter will come up later. DONE

**Specific Comment P.8 L.26-27:**

DICE is a globally aggregate model. While it would be interesting - indeed, highly important - to study the effect of inequality, this is outside the scope of the current, exploratory, study.

**Specific Comment P.9 L.1-6:**

This is really getting repetitive.

I agree it is helpful to clarify in the beginning and / or discussion that our model is limited and our study therefore only an exploratory one, (as we did now in the introduction); but it makes no sense to discuss the same things over and over and over again throughout the text (or the review).

Discounting is a different phenomenon from the duration/reversibility of damage. The way to take into account the irreversibility of loosing the Amazon rain forest would not be to fumble around with two discount rates, but to change the damage function such that this contribution to the damage does not decrease again even if temperature deceases again. Note that our tipping point obeys this type of irreversibility dynamics - once tipped, the associated loss remains constant (even if temperature should decrease again), as pointed out in the beginning of section 2.1.4.

**Specific Comment P.9 L.20-21:**

Reformulated (following also a suggestion by reviewer 2) as: "Although the objective for the optimisation is the expectation value of the welfare, it is also interesting to investigate the range of possible welfare outcomes, especially the worst (or at least relatively bad) case scenario." DONE

**Specific Comment P.9 L.29:**

Yes, in the sense that it was used as benchmark (see also result section), e.g. to compute performance. Clarified. DONE

**Specific Comment P.10 L.6:**

The idea of this scenario is again to be conservative and check whether SRM might make sense even under unfavourable conditions - e.g. taking a long time to develop the technology. The scenario serves as comparison to the Abatement+SRM scenario where no such restrictions were present.

As a matter of fact, while obviously the parameters (time till SRM becomes available and likelihood of becoming available) are arbitrary and not constrained by data (so yes of course, one could have taken 2045 instead of 2055), I do not think they are particularly implausible. Properly assessing risks and efficiency, developing the technology, developing a legal framework for international collaboration, stopping security leaks, and convincing the general public that SRM is a good idea, are all difficult processes with an uncertain outcome. Nearly all big (government) projects take much longer than planned. Many pilot project of CDR/CCS have been stranded or at least greatly delayed due to public discontent, technical difficulties, economic problems, and so on.

In addition, it is unclear whether sufficient radiative forcing can be generated (see Kleinschmitt et al, 2018 cited in the introduction).

Clarified motivation for the scenario at the end of section 2.3. DONE

On irreversibility of damages, see remark p.9 L.1-6

**Specific Comment P.10 L.14:**

I doubt whether the term you mention is bigger than the extra energy demand to build and fuel the SRM airplane fleet... or the change in need for warming in winter... or the effect of temperature on plant respiration and the solubility of $CO_2$ in the ocean. Most of these terms are quite uncertain. Without denying that SRM could influence $CO_2$ emissions and uptake, it makes little sense to try and include them into such a simple, exploratory study, especially not if we cannot even be sure whether we get the overall sign right.

Also, which study do you refer to? If it is the report by IEA: I just heard a presentation by a student of Guus Velders, mentioning that the study has considerable flaws.

**Specific Comment P.10 L.16:**

We did not include running-out of fossil fuels to avoid computational complications.

While the reserves of gas, oil and coal that we are relatively sure to be mineable will last a limited amount of time (about 50, 50 and 100 years, respectively; estimates from 2010-2014), it is believed that there are vast amounts of additional resources (e.g. fracked gas which since then came more and more into use; and potentially clathrates for gas), and undiscovered coal reserves. Some estimates assume that at present consumption rates, coal might last for 1000 years. So our no-action scenario is maybe extreme, but not impossible.

Source: Cassedy, E.S. and P.Z. Grossman, "Introduction to Energy", third edition, Cambridge University Press, 2017; p31-32.

We acknowledged this in the manuscript. DONE.

**Specific Comment P.10 L.19:**

To cool the earth to pre-industrial from, say, RCP8.5 radiative forcing in 2100 would involve sulphur injections amounting to something like *14 Pinatubo eruptions* every year. Surely, that would be disruptive, too...

The sentence is simply a description of results using the current damage function. You can argue that maybe the damages for warming should be higher and for sulphur lower, especially for situations with high CO2 and high SRM. This would lead to more SRM and lower temperature in the SRM-only scenarios. But even then, you would very likely see that, since SRM is so inefficient at high forcings, it is a bad idea to lower extremely high temperatures back to pre-industrial with SRM alone. It is way better to use also abatement, or, if this were not done, you would still resort to reducing some, but not all, warming with SRM (also because admitting some warming will reduce the precipitation-related damage which would result from compensating all warming by SRM).

So the main conclusion to draw from the SRM-only scenario is that if you care about long-term effects, you'd better do abatement. We pointed this out more clearly. DONE.

**Specific Comment P.11 fig.2:**

The constraint of SRM starting not before 2055 only holds for the scenario dubbed "realistic storyline" (for which the results are only discussed later, see fig. 4), not in the stylised scenarios of fig. 2. This is outlined in section 2.3.

Concerning the time axis: We usually omitted the first time step, as it is a spin-up step. Now provided a consistent figure. DONE

**Specific Comment P.12 tab.3:**

Indeed, this is the value in the first time step - we now point this out in the caption of table 2 and 3. DONE

For remarks about reliability of SCC, see p14, L25ff.

For the calculation of abatement costs, see the cited literature on the DICE model. As we now briefly mention in the introduction, DICE assumes abatement always to be costly, although abatement costs go down in time. This is certainly questionable; but here we wanted to explore qualitatively what happens when SRM is added, and not rewrite the entire DICE model at once...

See also p.19 lines 1-3 for a short remark on the issue of learning by doing.

**Specific Comment P.12 L.14-15:**

See numerous previous comments concerning tipping points

**Specific Comment P.13 fig. 3:**

Is this actually a question? Concerning abatement and energy transition, see remark P.12, tab.3

**Specific Comment P.14 L. 7:**

This is the SRM-only scenario, for which we did *not* assume SRM to start only in 2055. See Methods section (sect. 2.3).

All we state here is that (in fig. 3b) the policy maker increases SRM when hitting the 2K threshold, in order to reduce the chance of tipping.

**Specific Comment P.14 L. 16:**

this seems not to be a question or suggestion...

**Specific Comment P.15 fig. 4; P.16 L. 1:**

See numerous previous comments on damage function and general DICE problems.

The conclusion from "realistic storyline scenario" vs. "Abatement+SRM" (which assumes immediate availability of SRM at 100% likelihood) is: As soon as SRM becomes available, if at all, it should be used to complement Abatement - unless termination shock damage and failure probability are really high.

This result would not change under a sensitivity study as suggested here.

**Specific Comment P.16 L. 2-3; L. 5-6:**

Clarified in Methods section on failure that it is indeed irrevocable. Also clarified there that failure is at present state speculative (though not impossible).

The sensitivity run on $\kappa_{fail}$ showed that the main results are not very sensitive to the exact failing probability (if SRM is available, it is used even if it might fail somewhen).

**Specific Comment P.16 L. 17:**

I cannot see a connection between this remark and P16, L17, sorry.

abatement costs have been discussed already in previous comments....

**Specific Comment P.16 L. 26:**

Discussion on reliability of absolute CCS values: See p.14, l.25, including the reference to van den Bergh and Botzen cited there, which explains why CCS should be higher than DICE-like models.

**Specific Comment P.16 L. 31-33:**

Do we really need to estimate/conjecture/contextualise "huge" damages for an effect which we did *not* take into account? All we are saying here is that we *do not include scenarios* in which research on SRM (which would probably include testing at some stage) will cause huge/significant/whatever damages.

Of course, research cost on SRM is one of the myriad things one could investigate. But this would really blow up computational costs due to having another variable - especially if you want this to be an extra decision variable -, it would also introduce yet another layer of ill-constrained parameters (likelihood and size of damage during tests) and the effect on the results would likely be small. The possibility of "something going wrong early on with SRM" is qualitatively included by having a high failure probability in the first steps after SRM becomes available.

**Specific Comment P.17, L1:**

What DICE really needs is a proper energy sector, including effects like learning-by-doing, and that is outside the scope of this study.

We did, however, perform a sensitivity study where abatement costs remain exogenous, but decrease faster than in the standard settings, to investigate whether abatement costs have a large influence on policy (and thus whether this is an important aspect to study with better models). The result is that if abatement costs increase faster, more abatement and less SRM is used. So indeed abatement costs (and hence probably learning-by-doing) affect the need for SRM.

DONE.

**Specific Comment P.17, L9:**

If you do more abatement, you need less SRM.

CO2 has a long residence time; so abatement done now also affects future carbon levels and therefore reduces the need for SRM. SRM will always come at costs, so for a future generation it is always beneficial of they have to do less of it because a previous generation has done more abatement for them.

The lower $\rho$, the more prepared the earlier generations are to pay for abatement.

By "SRM causes damage" it is meant that SRM makes a contribution to damage, see $\psi_S$.

**Specific Comment P.18, L3:**

Nice suggestion. Result: SRM increases by ca 25%, but no qualitative change in policy. See section on sensitivity runs.

DONE

**Specific Comment P.18, L21 ff:**

As mentioned, I don't believe in constantly repeating things. Scientific papers are meant to be concise.

We have already put a cautioning remark on the most prominent DICe problems in the introduction, and expanded the section in the discussion on how these problems may affect our results, and repeated that CDR and adaptation are not included (even though tradeoff between SRM and adaptation and CDR would be important to study in folow-up research).

We also want to do an additional sensitivity study on abatement costs (See answer to comment p17, l1), to get a better idea of how large the influence of this effect on the policy. DONE

**Specific Comment P.19, L5-7**

Concerning what is / is not included in damage function, see numerous previous comments.

Added "...uncertainties, especially concerning efficiency and damages of SRM and *the extent by which SRM can mitigate damage inflicted by global warming*" DONE

**Specific Comment P.20, L.30 ff**

Interesting remark, although not of practical importance here, because throughout all simulations, no situation occurs in which the planet is too cool. As a side remark, adding CFCs or HFCs would not be exactly reverse SRM, because for example the effect on precipitation would be different.

**Technical comments**

- P1L1: we keep in spite of, to stress that (unlike abatement or CDR) SRM has impact even though greenhouse gas concentration remains high.

- P1L8: no "can" found in this line. changed "cannot replace but only complement" to *"can not* replace but only complement" to elucidate the structure of the sentence.

- P1L14: I see your point, but the reference was more to the geoengineering as such, not to the history of the discussion of geoengineering.

- P1L20: the connection with volcanic eruption is made just 1 sentence later, so I'd prefer to not make this sentence overly clumsy.

- P1L21: thanks, corrected.

- P2L4: reformulate for clarification

- P2L23-24: Moreno-Cruz corrected. that: corrected

- P8L2: refers to Heutel's parametrisation for CO2-related damage (not ours). Rephrased.

- P10L18: indeed, SO2 is injected (and leads to sulphate formation). Clarified.

- P12L10: corrected

- P13Fig3: time axis: because we were interested in how tipping continues in the course of time. For the other plots we zoomed in on the first ca 300 years for better readability, because nothing interesting happens in the last 100 years. (will clarify)
  We prefer not having the same y-axis for the mentioned graphs because then some graphs become very hard to read, especially SRM plots in simulations with abatement.
  Spelling: thanks, will be corrected. DONE

- P14L8: performance is explained in eq. 11

- P14L9: the simulation goes to year 2400 even though the graphs do not (see remark above). CDR is not modelled, as you know.

- P20L21: this means that for "safety", the actual margin is made wider than the one suggested by the test trajectories, namely by adding 30% to the maximum and subtracting 30% from the minimum.

- P22ff: Stanford: corrected; title case: sorry, what is meant by this?; ncomms=nature communications (journal); Stowe: corrected.

**Reply to Reviewer 2**

**June 6, 2019**

*First, a more interesting question about SRM is, to me, not the precise level at which we should be using it now but whether we should start using it at all. The current model has no force that would lead a policymaker to delay introducing SRM. I can think of two relevant forces. First, there may be a fixed cost to beginning SRM, whether political, economic, or ecological. Second, the authors assume (I think) that the risk of SRM failure is constant over time. A more realistic model would have a small chance of new research finding a fatal flaw in SRM as long as it is not used and a much larger chance of actual consequences revealing a fatal flaw once it is used. I am suggesting that the probability of failure should be low as long as SRM is not used and then be higher once it is used. A policymaker might then choose to delay beginning SRM until it is really needed, since SRM may not last for long once it is begun. I strongly encourage the authors to explore these (or other) ways of making the paper about when to begin using SRM.*

While in the standard scenarios (section 3.2) we indeed assumed constant failure probabilities for simplicity, the failure probability is time dependent in the "realistic storyline" (Section 3.3), although it is still exogenous, i.e. independent of whether SRM is used yet.

Unfortunately, in the dynamic programming procedure, very high additional computational costs are incurred for adding another state variable, for example for tracking the history of whether and when SRM has started. This makes it very cumbersome to include costs depending on how long ago SRM started (e.g. an initial cost to be payed when starting SRM).

As a compromise, we have run a set of simulations with various (high) costs in case of SRM failure, similar to the termination shock damage in section 3.3. The aim is to investigate whether the fear of very large risk associated with SRM failure might lead to delay (of strong decrease) of deployment. SRM is suppressed until the climate tipping threshold of 2K is reached. See new sect. 3.4.

    DONE

*Second, the interesting aspect of SRM value is, to me, its insurance value. The authors analyze a dynamic programming problem, so they have the machinery to answer that kind of question, but they fail to get there. Instead, they analyze perhaps the least interesting form of "tipping" that they could have. Their tipping point permanently reduces economic output, but the interesting aspect of SRM in a world with tipping points is the potential for SRM to manage a tipping process that is underway. Had the authors chosen to model tipping points in the way that Lemoine and Traeger (2014) did (where a tipping point is a sudden change in a parameter of the physical system, which manifests itself in temperature and economic output only over time and in a fashion that depends on subsequent policy), SRM could play a key role by allowing the policymaker to intervene after triggering a tipping point so as to mitigate the consequences of that tipping point. Alternately, had the authors studied uncertainty about future warming, SRM could have been used to control the consequences of ending up in a high-warming world. Finally, if the model allowed for the more interesting type of tipping point (or for warming uncertainty) along with a reason to delay beginning SRM, then we may see the policymaker delay SRM until a bad state of the world warrants using it. Learning whether/when that plan is optimal would be interesting..*
*https://are.berkeley.edu/~traeger/pdf/Lemoine%20Traeger_Watch%20your%20Step_AcceptedAEJPolicy.pdf*

This alternative climate tipping point is a very interesting suggestion, thank you very much.

We have added a simulation based on the Abatement+SRM where we replaced the damage tipping point by a "sudden warming" tipping point. If this tipping point is triggered, an additional positive radiative forcing contribution $F_{alb}$ will be added to $F$ (which can be thought of as an unexpectedly strong albedo feedback).

The additional forcing obeys:

$$F_{alb} = \alpha_{alb} \max(T - T_{alb}, 0) \tag{1}$$

and the tipping probability obeys the same equation as for the damage tipping point, except that $T_{alb} = 1.5K$ is chosen as threshold.

Note that this tipping point is reversible in the sense that $F_{alb}$ can decrease again if $T$ decreases.[1]

The result is, interestingly, that if he may use SRM, the policy maker does not try to avoid this tipping point. Rather, in case the tipping point is crossed, he reacts by increasing SRM afterwards such as to offset the warming induced by this (reversible) tipping point.

The results are described in more detail in the new section 3.4.

For "delaying" SRM, see answer to the previous comment. In the scenario with the albedo tipping, SRM is increased in case of hitting the tipping point, but it is already used before.

DONE

**SMALLER COMMENTS**

*The authors should spend more space clearly laying out the methodological, calibrational, and conceptual contrasts with Heutel et al. They also should elaborate on the differences in results and speculate as to their origin,*

Main Methodological differences between our study and Heutel 2018:

- Their optimisation scheme is a 4-day look-ahead scheme, which is not suitable for long-term optimisation.

- They use a different technique to take into account uncertainty of "how harmful unmitigated CO2 is" and how damaging SRM is, namely by assuming the policy maker to be uncertain about climate sensitivity and SRM damage (without learning). In contrast, we use tipping and SRM failure. (Incorporating parameter uncertainty would be very computationally demanding using dynamic programming.)

- Their climate-related damage function is split in a different fashion, ignoring residual climate change (they do include direct effects from CO2, though).

- SRM implementation costs and damages are both implemented in an implausible fashion, namely as being proportional to the *fraction* of CO2-induced radiative forcing removed by SRM. I.e. it does not matter whether the CO2 forcing is strong or weak; the damage induced for compensating X % of this forcing remains the same. In particular, the fact that a high CO2 content requires absurdly high SO2 emission rates (making a full compensation of global warming undesirable) is not included; the maximal damage from SRM is 3% of GDP (full compensation of CO2-induced forcing); we have no such limit.

The effects of the first item onto the results are hard to predict - their optimisation scheme *might* yield unreliable results, but we cannot tell a priori if this will have a strong impact and to which direction.

The second item: Uncertainty in the "harmfulness of CO2" (climate sensitivity in Heutel; tipping in our study) has a larger policy impact in Heutel, at least for the Abatement+SRM-like scenarios in absence of SRM failure, because in our study, Abatement+SRM keeps below the tipping threshold.

The third item would - at least if the SRM-related damage and implementation costs were equal - probably lead to more SRM in Heutel, because SRM always diminishes climate-related damage (namely, the 80% associated with temperature). In our results, even if SRM caused no damage or cost on its own, it would be optimal to compromise between temperature-related and rainfall-related (residual) damage. However, this might be a minor effect.

The last item causes the most obvious differences. In Heutel's case, the higher the CO2 concentrations (and thus the more global warming), the higher the fraction of greenhouse gas forcing that on wants to balance by SRM, as one deletes more global warming damage for the same SRM-induced cost. So Heutel should be more (less) inclined to rely on SRM for high (low) CO2 concentrations, compared to our model.

It seems that in their deterministic results (Heutel et al., 2018; fig. 2), their peak SRM happens to be somewhat similar to ours: In their year 2140, the highest atmospheric CO2 content is about 1800GtC, leading to a forcing of about 6W/m^2, of which they compensate about 50%. In our model, compensating 3W/m^2 of radiative forcing requires 27Mt(S)/yr; similar to our peak abatement of 35Mt(S)/yr (which we reach later than 2160, namely around 2220; see our fig. 2b). However, around 2020, Heutel et al. compensate 10% of a radiative forcing of 1.6W/m^2,
* * *
[1]This is why it makes actually more sense to think of this stylised tipping point as related to albedo, rather than methane release from permafrost, as we suggested in the first version of the letter to the reviewer. Released methane would stay in the atmosphere for some time. However, for computational reasons we did not want to introduce another time-dependent variable for the methane stock.

which is equivalent to about 1.5Mt(S)/yr in our model. In our results, the injection rate around 2020 is almost 10W/m^2 (compensating 1.5W/m^2 of radiative forcing). So basically, especially during the first 100 years or so, while CO2 concentration is not yet so high, Heutel et al. use considerably less SRM, because they overestimate its cost (i.e. ignore that for low CO2 concentrations, there is relatively little radiative forcing that needs compensation). This might explain why they get higher temperatures than we (see their fig. 2d and ours).

Although they do not explore it, if Heutel et al. had used some high-emission scenario like SRM-only, they would unrealistically find that they can always eliminate 80% of their climate damage (the temperature contribution) for a max. cost of around 3%GDP (max. SRM damage), which would imply that SRM is a very good emergency technology (in case that for some reason abatement is not working or in case of unexpected CO2 release from permafrost).

We will include an abbreviated version of this comparison (focusing on the fourth methodological difference) into the discussion section of our paper.
DONE

*-Technical notes: 1) The authors describe how they mitigate the problem being non-smooth, but it seems far more straightforward and more realistic to simply assume that the chance of tipping is smooth. Assuming no risk of tipping below some temperature is arbitrary.*
*2) The authors impose an upper bound on injection rates. This is a curious choice. I would prefer the authors to model the source of the constraint. As it is, either the constraint binds or it does not. If it binds, then it is important, the authors should highlight it in their results discussion, and the authors should consider eliminating it. If it does not bind, then it is unimportant and the authors should consider eliminating it (or at worst justify it in terms of numerical convenience but say it is irrelevant in practice)*

1) Any choice of ill-constrained parameters is inherently arbitrary. Ours was inspired by the Paris agreement of staying "well below" 2K warming, because additional warming would be "too dangerous". We could of course have picked a different threshold, such as 1.5K or 0K, or a quadratic increase of tipping probability with temperature. Note that although the 2K (or maybe 1.5K) threshold may be hard to justify on physical grounds, it does match the current political approach of trying to avoid crossing a certain temperature threshold.

We have performed a sensitivity run using 1K as threshold in the standard Abatement+SRM scenario. The effects on policy are very small (see section on sensitivity runs).
DONE
2) Yes, indeed, a threshold was needed for numerical reasons. It has no practical consequences except towards the end of the SRM-only run (which is a somewhat unrealistic scenario anyway). We will clarify this in the manuscript.
DONE

*Sensitivity checks: I want to see more sensitivity analysis with respect to the fairly arbitrary damage parameters, and especially to the $\psi S$ that I'm not sure has been explored. This should be the core of the results. I am far more interested in learning about what we need to believe to get near-term SRM to be high or low than in learning about the spread of future policy.*

We already have a sensitivity run with $\psi_S$ doubled; this leads to a slightly faster abatement (6 years) and lower peak SRM (ca 23% reduction), but no qualitative change in policy. We now also included a simulation with $\psi_S$ halved. Again, there in no qualitative change in the policy (SRM increases by 25% and Abatement is slightly delayed).
DONE

*-I wasn't happy with the discussion of precipitation damages in 2.1.3. There are plenty of ways precipitation can matter, independently of temperature. Think of crops. One could have just as easily said that many temperature channels are mediated by precipitation. Plus geoengineering changes precipitation in ways that are not merely determined by temperature. For instance, might geoengineering change patterns of rainfall (such as monsoons)? Perhaps these kinds of effects could be captured by $\psi S$, about which very little is said other than that its level is arbitrary.*

I do not fully understand this remark. We did assume that precipitation can matter independently of temperature, that's why we split the damage function into a precipitation and a temperature component (which are

differently affected by SRM).

It was mentioned at the end of 2.1.1 that global mean precipitation changes serve as proxy for the residual climate change - similar to the way that Nordhaus uses temperature change as an indicator for "climate change", including changes which are not directly caused by temperature (e.g. rainfall changes). Nordhaus needed only one climate indicator, because he could assume that all climate change somehow scales with temperature, while we must differentiate between climate change that can / can not be mitigated by SRM.

An interesting aspect about precipitation change is that SRM and CO2 have opposing effects and can balance each other. Therefore even if SRM caused no damages or costs on its own, it might be beneficial to compromise between reducing warming and precipitation changes. This interplay would be lost if precipitation damages were simply included into $\psi_S$.

We will add a remark for clarification. DONE

*-The objective in (10) must be wrong. I think the policymaker must be maximizing expected welfare, not just welfare. And in light of that comment, the motivation for the final paragraph in 2.2 doesn't make a lot of sense. A risk-averse policymaker (as modeled here) has already chosen policy in light of the range of possible cases. It is not internally consistent to postulate caring about percentiles of the subsequent policy performance because of risk aversion.*

On eq. 10: indeed, in the stochastic case one must optimise *expected* welfare. We corrected this in the text above the equation ($W$ defined in eq.10 is supposed to be the welfare of a specific realisation, and the expectation value of $W$ is to be optimised. DONE

Concerning the motivation of the final paragraph of 2.2: I do not fully agree. Even if one chooses expected welfare as objective for the optimisation, it is interesting to at least look at other criteria such as the percentiles used here (for example, a policy maker might still want to know the probability of "something going quite wrong" under a certain policy). In particular for the realistic storyline scenario, in which SRM is only available with certain probability, it is also insightful to consider not only the mean but include more measures (to verify that the increase in expected welfare compared to Abatement-only comes from those simulations where SRM became available).

We will clarify in the text that the additional measures are not used as optimisation objectves but for additional information only. DONE

*-In s 3.2, the authors report that abatement decreases after triggering a tipping point. This is purely an artifact of the type of tipping point modeled here. It would probably not arise if abatement had any role in controlling the consequences of a tipping point, as in Lemoine and Traeger 2014.*

Indeed, with the albedo tipping point (see major comment 2), abatement goes up, not down, after hitting the tipping point. We will point this out, also in the context of the new albedo tipping point. DONE

*-Why is the scc unaffected by the possibility of tipping in the abatement+SRM world?*

There might be a small effect which however is so small that it vanished when rounding to whole dollars. The reason why this effect is small is that tipping is quite unlikely in this scenario, especially at the early (less discounted) time steps, because unless SRM fails, the temperature is kept below the tipping threshold.

*-It seems to me that the final sentence on page 16 would be stronger with a comparison between maximized welfare in the "realistic" world and welfare in the same world if the policymaker used the policy from the abatement-only world. How much could today's policymaker gain by anticipating the future possibility of SRM?*

That very last sentence on p16 was meant not so much to stress the potential gain, but to warn against slackening abatement while SRM is uncertain. We will stress your point in earlier in the paragraph and reformulate the last sentence to make this more clear. In particular, we will clarify that for the "realistic" scenario, there is a welfare gain by almost 200% in those cases where SRM becomes available (compared to the "abatement-only" case). DONE

*-The discounting result in 3.4 was one of the more interesting results of the paper: a higher discount rate favors SRM because it causes damages, may fail, and doesn't curb future CO2. I would suggest highlighting it more and perhaps doing more with it.*

We will highlight the result by mentioning it in the conclusions (to further stress that abatement is indispensable, especially if one "cares much about the future", while SRM is more a short-term measure). DONE

**List of substantial changes**

June 6, 2019

New simulations

- Added two additional scenarios on SRM as insurance (see new section 3.4, also P10L1-9, and )

- Added sensitivity studies: faster decline of abatement cost; reduced damage from SRM; lower tipping threshold. See current section 3.5, particularly table 4.

Major textual issues (clarifications, corections, etc):

- P2L13: added comparison SRM cost - estimate for abatement cost

- P2L29ff: added explanation of major shortcomings of DICE model

- P3L11-12: clarified codes on which we built

- P3L18: corrected error (missing exponent $\gamma_{SO2}$) in eq.1 (not pointed out by reviewer)

- P3L27: added troposphreic ozone as important non-CO2 greenhouse gas

- P4: Added some parameters to the table, line 20-23 and 25-27

- P5L19-23: clarified that limit on SRM is due to numerical constraints but of little importance in nearly all simulations

- P6L22ff: better explained precipitation response

- P7L5-7: added clarification about splitting precip. response into temperature-driven and fast response

- P7L20-24: added equation on abatement cost in order to provide context for the new sensitivity run with faster decrease of costs.

- P7L29-31: clarified why we assumed SO2 and not H2S to be used for SRM

- P8L6-11: Improved explanation concerning splitting the damage function

- P9L21-23: Justfied choice of 2K as tipping threshold

- P9L26-28: Justified assumption that SRM might be abandoned

- P9L31-32: clarified that econ. damage associated with SRM failure does occur in some of the scenarios

- P10L1ff: explained albedo tipping point (See sect. 3.4)

- P10L14: corrected ambiguity: the quantity to be optimised is the expectation value of the welfare (and welfare is defined in eq. 12)

- P11L4-5: clarified that expectation value of the welfare (and not percentiles considered below) is the quantity to be optimised.

- P11L17-18: menioned that no-policy scenario is also computed (as benchmark)

- P11L21-30: justified delay of SRM availability in Realistic Storyline scenario and added clarifications, in particular on termination shock.

- P13L16-18: pointed out that SRM might serve as transition technology to "shave off" a warming peak temporarily until abatement shows sufficient effect

- P13L26: clarified that abatement is needed to stabilitse temperature in the long run

- P14L7-8: pointed out that reduction in abatement after hitting the tipping point is to some extent caused by our implementation of tipping points.

- P18 last paragraph: clarified that better performance of "realistic Storyline" vs "Abatement only" is due to those ensemble members where SRM is made available. stressed more clearly that abatement should not be strongly reduced at current stage while SRM is still quite uncertain.

- sect.3.4 and fig. 5: new scenarios on insurance behaviour mentioned in beginning of this list.

- p20, last full paragraph: senitivity run with quickly-decreasing abatement costs

- p21, text and table 5: added sensitivity studdies with lower tipping threshold and lower SRM damage (see beginning of this list)

- p22, first paragraph of sect.4: pointed out impact of reduced rate of pure time preference and possible role of SRM as "insurance"

- P22 second last paragraph: added comparison with study by Heutel

- P23 second last paragraph: expanded explanation of how shortcomings of DICE might affect results

For the minor changes, such as typos, small reformulations, added references to equations within table 3, etc, please consult the change track document.

[revised manuscript text omitted]

35 Myhre, G., D. Shindell, F.-M. Bréon, W. Collins, J. Fuglestvedt, J. Huang, D. Koch, J.-F. Lamarque, D. Lee, B. Mendoza, T. Nakajima, A. Robock, G. Stephens, T. Takemura and H. Zhang, 2013: Anthropogenic and Natural Radiative Forcing. In: Climate Change 2013: The Physical Science Basis. Contribution of Working Group I to the Fifth Assessment Report of the Intergovernmental Panel on Climate

Change [Stocker, T.F., D. Qin, G.-K. Plattner, M. Tignor, S.K. Allen, J. Boschung, A. Nauels, Y. Xia, V. Bex and P.M. Midgley (eds.)]. Cambridge University Press, Cambridge, United Kingdom and New York, NY, USA.

Niemeier, U. & Timmreck, C. What is the limit of climate engineering by stratospheric injection of $SO_2$? *Atmos. Chem. Phys.,* 15, 9129–9141 (2015)

5    Niemeier, U. & Schmidt, H. Changing transportprocesses in the stratosphere by radiative heating of sulphate aerosols.*Atmos. Chem. Phys.*, 17, 14871–14886, (2017)

Nordhaus, W. D. The "DICE" Model: Background and Structure of a Dynamic Integrated Climate–Economy Model of the Economics of Global Warming. Cowles Foundation Discussion Paper No. 1009. Cowles Foundation for Research in Economics: New Haven, CT (1992)

Nordhaus, W.D. and J. Boyer, Warming the World - Economic models of Global Warming, The MIT Press (2000)

10   Nordhaus, W.D. Evolution of Modeling of the Economics of Global Warming: Changes in the DICE model, 1992-2017, Climate Change 148 (4): 623–40. (2018)

Pindyck, R.S. The Use and Misuse of Models for Climate Policy, Review of Environmental Economics and Policy, 11, 100-114 (2017)

Pitari, G., et al. Stratospheric ozone response to sulphate geoengineering: Results from the Geoengineering Model Intercomparison Project (GeoMIP), J. Geophys. Res. Atmos., 119, 2629–2653 (2014)

15   Robock, A. Volcanic eruptions and climate. Reviews of Geophysics, 38, 191–219 (2000)

Robock, A., Marquardt, A., Kravitz, B., & Stenchikov, G. Benefits, risks, and costs of stratospheric geoengineering. Geophys Res Lett 2009, 36:L19703 (2009)

Seneviratne, S.I. et al. Land radiative management as contributor to regional-scale climate adaptation and mitigation. Nat. Geosci. 11: 88–96 (2018)

20

Stenchikov, G. L., et al. Radiative forcing from the 1991 Mount Pinatubo volcanic eruption, J. Geophys. Res., 103, 13837–13857 (1998)

Stern, V. et al. The Stern Review. Government Equalities Office, Home Office (2007)

Stowe, L.L., R.M. Carey, P.P. Pellegrino, Monitoring the Mt. Pinatubo aerosol layer with NOAA/11 AVRHH DATA, Geophys. Res. Lett. 25   19/2 159 (1992)

Thompson, W. J., Wallace, J. M., Jones, P. D. & Kennedy, J. J. Identifying Signatures of Natural Climate Variability in Time Series of Global-Mean Surface Temperature: Methodology and Insights J. Clim. 22, 6120–6141 (2009)

Tilmes, S. et al. CESM1(WACCM) Stratospheric Aerosol Geoengineering Large Ensemble (GLENS) project. Bull. Am. Meteorol. Soc. (2018).

30   Tjiputra, J. F., Grini, A. & Lee, H. Impact of idealized future stratospheric aerosol injection on the large-scale ocean and land carbon cycles, J. Geophys. Res. Biogeosci. 121, 2–27 (2016).

Trisos, C. H. et al. Potentially dangerous consequences for biodiversity of solar geoengineering implementation and termination. Nat. Ecol. Evol. 2, 475–482 (2018)

UNFCCC(2015), Adoption of the Paris Agreement, United Nations Framework Convention on Climate Change, United Nations Office, 35   Geneva, Switzerland.

J.C.J.M. Van den Bergh, W.J.W. Botzen Monetary valuation of the social cost of CO2 emissions: a critical survey Ecol. Econ., 114, pp. 33-46 (2015)

Visioni, D., Pitari, G., & Aquila, V. Sulfate geoengineering: A review of the factors controlling the needed injection of sulfur dioxide. Atmospheric Chemistry and Physics, 17(6), 3879–3889 (2017)

Ward 2009: sulphur dioxide initiates global climate change in four ways Thin Solid Films 517, 3188–3203 (2009)

World Bank (https://data.worldbank.org/indicator/NY.GDP.MKTP.CD)